# Clustering Properties of Self-Supervised Learning

**Xi Weng** [1 2]    **Jianing An** [1]    **Xudong Ma** [1]    **Binhang Qi** [2]
**Jie Luo** [1]    **Xi Yang** [3]    **Jin Song Dong** [2]    **Lei Huang** [✉ 1 4]

## Abstract

Self-supervised learning (SSL) methods via joint embedding architectures have proven remarkably effective at capturing semantically rich representations with strong clustering properties, magically in the absence of label supervision. Despite this, few of them have explored leveraging these untapped properties to improve themselves. In this paper, we provide an evidence through various metrics that the encoder's output *encoding* exhibits superior and more stable clustering properties compared to other components. Building on this insight, we propose a novel positive-feedback SSL method, termed **Re**presentation **S**elf-**A**ssignment (ReSA), which leverages the model's clustering properties to promote learning in a self-guided manner. Extensive experiments on standard SSL benchmarks reveal that models pretrained with ReSA outperform other state-of-the-art SSL methods by a significant margin. Finally, we analyze how ReSA facilitates better clustering properties, demonstrating that it effectively enhances clustering performance at both fine-grained and coarse-grained levels, shaping representations that are inherently more structured and semantically meaningful.

## 1. Introduction

Self-supervised learning (SSL) has emerged as a transformative paradigm in universal representation learning (Oord et al., 2018; Bachman et al., 2019; He et al., 2020; Chen et al., 2020a; Bao et al., 2021; Oquab et al., 2023; Assran et al., 2023), consistently surpassing supervised learning in downstream performance. Joint embedding architectures (JEA), in particular, aim to learn invariance of the same data

[1]SKLCCSE, School of Artificial Intelligence, Beihang University [2]School of Computing, National University of Singapore [3]Beijing Academy of Artificial Intelligence [4]Hangzhou International Innovation Institute, Beihang University. Correspondence to: Lei Huang <huangleiAI@buaa.edu.cn>. Our code is available at https://github.com/winci-ai/resa

*Proceedings of the 42nd International Conference on Machine Learning*, Vancouver, Canada. PMLR 267, 2025. Copyright 2025 by the author(s).

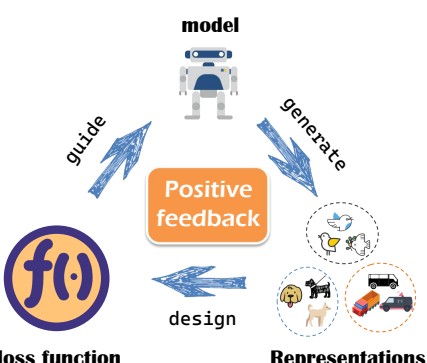

*Figure 1.* The positive-feedback SSL framework. It involves the model generating representations that possess semantically clustering information. This clustering information is leveraged to design self-supervised loss function, which is then employed to more effectively guide the model's learning process.

under different transformations and noise (Bachman et al., 2019; He et al., 2020; Chen et al., 2020a), with demonstrated exceptional effectiveness in visual representation learning. Although such a pretext task may intuitively seem unrelated to capturing semantic relationships, extensive studies (Caron et al., 2018; 2021; Assran et al., 2022b) have demonstrated the strong correlation between its learned representations and semantic information.

Ben-Shaul et al. (2023) take a further step to characterize the semantic structures learned by JEA into hierarchic clustering properties, indicating SSL-trained representations exhibit a centroid-like geometric structure and induce three levels of semantic clustering: augmentation sample level, semantic classes, and superclass level. This intriguing finding reveals that SSL methods based on JEA can facilitate strong clustering capabilities during training, but also raises the question of *whether these properties hold untapped potential that can be further leveraged to improve SSL itself*.

**Contributions.** In this paper, we aim to investigate the design of SSL methods by leveraging the inherent clustering properties of representations, enabling a closed-loop positive-feedback SSL framework, as illustrated in Figure 1. To achieve this goal, we propose three key questions and our main contributions are summarized as follows:

- **Where to extract clustering properties from?** We demonstrate through various metrics that the encoder's output, referred to as *encoding*, exhibits superior and more stable clustering properties compared to other

components, such as *embedding* and the *hidden layer outputs* within the projector.

- **How to leverage clustering properties?** We propose a novel SSL method, termed **Re**presentation **S**elf-**A**ssignment (ReSA), which employs an online self-clustering mechanism to leverage the model's inherent clustering properties, thereby facilitating positive-feedback learning. Standard experiments demonstrate that ReSA surpasses existing state-of-the-art methods in both performance and training efficiency.

- **Whether it facilitates better clustering properties?** We examine whether and how ReSA facilitates better clustering properties, demonstrating that it excels at both fine-grained and coarse-grained learning, shaping representations that are inherently more structured and semantically meaningful.

## 2. Background, Related Work, & Notation

### 2.1. Self-Supervised Learning

$\mathbf{H}$: *encoding*   $\mathbf{Z}$: *embedding*

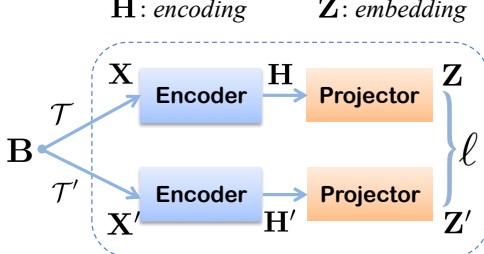

*Figure 2.* The basic notations for joint embedding architectures (JEA) in SSL.

**Joint embedding architectures (JEA).** Let $\mathbf{B}$ denote a mini-batch input sampled uniformly from a set of images $\mathbb{D}$, and $\mathbb{T}$ denote the set of data transformations available for augmentation. We consider a pair of neural networks $F_\theta$ and $F'_{\theta'}$, parameterized by $\theta$ and $\theta'$ respectively. They take as input two randomly augmented views, $\mathbf{X} = \mathcal{T}(\mathbf{B})$ and $\mathbf{X}' = \mathcal{T}'(\mathbf{B})$, where $\mathcal{T}, \mathcal{T}' \in \mathbb{T}$; and they output the *embeddings* $\mathbf{Z} = F_\theta(\mathbf{X})$ and $\mathbf{Z}' = F'_{\theta'}(\mathbf{X}')$. The networks are trained with an objective function that minimizes the distances between *embeddings* obtained from different (two) views of the same images:

$$\mathcal{L}(\mathbf{B}, \theta) = \mathbb{E}_{\mathbf{B}\sim\mathbb{D},\ \mathcal{T},\mathcal{T}'\sim\mathbb{T}}\ \ell\big(F_\theta(\mathcal{T}(\mathbf{B})), F'_{\theta'}(\mathcal{T}'(\mathbf{B}))\big). \quad (1)$$

where $\ell(\cdot, \cdot)$ is a loss function, which aims to learn invariance of data transformations.

In particular, the JEA usually consist of a shared encoder $E_{\theta_e}(\cdot)$ and projector $G_{\theta_g}(\cdot)$, commonly referred to as a Siamese Network (Chen & He, 2021). As shown in Figure 2, their outputs $\mathbf{H} = E_{\theta_e}(\mathbf{X}) \in \mathbb{R}^{d_e \times m}$ and $\mathbf{Z} = G_{\theta_g}(\mathbf{H}) \in \mathbb{R}^{d_g \times m}$ are referred to as *encoding* and *embedding*, respectively (where $m$ is the mini-batch size, $d_e$ and $d_g$ are their corresponding feature dimensions). Under this notation,

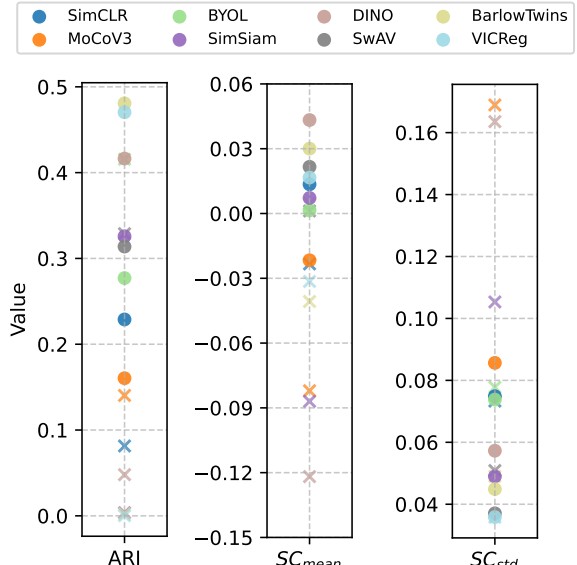

*Figure 3.* Comparison of clustering metrics in *encoding* $\mathbf{H}$ and *embedding* $\mathbf{Z}$ across various self-supervised pretrained models. All methods utilize a ResNet-18 encoder pretrained on CIFAR-10 for 1000 epochs. Circular markers represent metrics computed using *encodings*, while cross markers correspond to metrics derived from *embeddings*. All metrics are computed on the entire training set, and similar trends can be observed in the validation set.

we have $F_\theta(\cdot) = G_{\theta_g}(E_{\theta_e}(\cdot))$ with learnable parameters $\theta = \{\theta_e, \theta_g\}$. It is worth noting that extensive work (Chen et al., 2020a; Gupta et al., 2022) has demonstrated that using the *encoding* as the representation for downstream tasks achieves much better performance than using *embedding*, so that the projector is only used during the pre-training process and discarded in inference.

**Self-supervised paradigms.** The main challenge with JEA is representation collapse, where both branches produce identical and constant outputs regardless of the inputs. Numerous paradigms have been proposed to avoid collapse, including contrastive learning methods (Chen et al., 2020a; He et al., 2020; Chen et al., 2020b; 2021; Liu et al., 2022) that attract different views from the same image (positive pairs) while pushing apart different images (negative pairs), and non-contrastive approaches (Grill et al., 2020; Caron et al., 2020; 2021; Weng et al., 2024) which directly align positive targets without incorporating negative pairs. Although Ben-Shaul et al. (2023) had demonstrated that the *encodings* learned through SSL are highly correlated with semantic classes and exhibit strong clustering capabilities, few methods have leveraged this clustering ability to facilitate positive-feedback learning. A closely related work (Ma et al., 2023) exploits the *encoding*'s augmentation robustness to re-weight the positive alignment in the SSL objective functions; however, it still overlooks the rich clustering information inherent in the *encodings*.

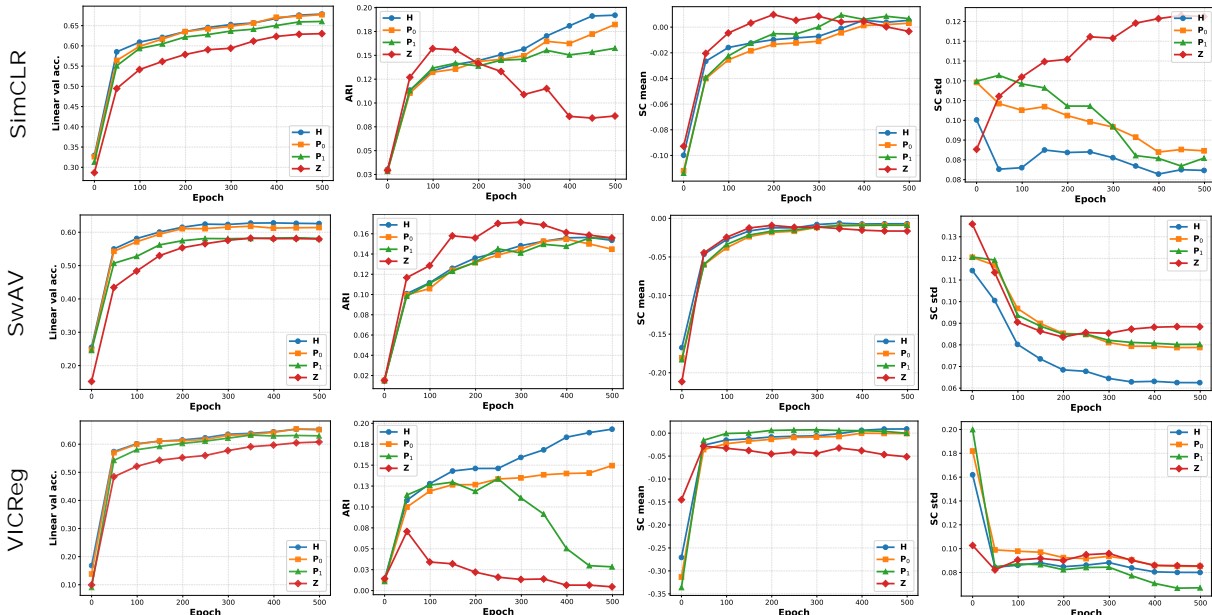

*Figure 4.* Comparison of linear evaluation accuracy and clustering metrics of *encoding* **H**, *embedding* **Z**, and the *hidden layer outputs within the projector* **P** during the training process. The experiments are conducted using SimCLR, VICReg, and SwAV, employing a ResNet-18 encoder pretrained on CIFAR-100 for 500 epochs. The projector is a standard three-layer MLP with BN and ReLU activations, containing two hidden linear layers, so their outputs are denoted as $P_0$ and $P_1$.

## 2.2. The Information Distinction in JEA

The projector has become an indispensable component of JEA-based SSL. However, the theoretical dynamics of its optimization and the reasons behind its success remain an open question within the community. Some works have attempted to explain these principles. For example, Jing et al. (2022) empirically discovered that applying SSL loss either to *encoding* or *embedding* led to a significant decrease in the rank of the corresponding features. They argued that this rank reduction indicates a loss of diverse information, which, in turn, reduces generalization capability. This explanation aligns with the hypothesis in SimCLR (Chen et al., 2020a), where the additional projector acts as a buffer to prevent information degradation of the *encoding* caused by the invariance constraint. Additionally, Gupta et al. (2022)'s null space analysis for the projector posited that the projector might implicitly learn to select a subspace of the *encoding*, which is then mapped into the *embedding*. In this way, only a subspace of the *encoding* is encouraged to be style-invariant, while the other subspace can retain more useful information.

Therefore, the SSL constraint can cause the *embedding* to lose information, which may include not only clustering-irrelevant features such as background information, but also class-relevant information, making it difficult to determine which—*encoding* or *embedding*—exhibits better clustering performance in this context. In such cases, this paper first analyzes the differences between the two in terms of clustering properties empirically.

## 3. Exploring Clustering Properties of SSL

Ben-Shaul et al. (2023) find that within the encoder of JEA, the clustering ability of features improves progressively as intermediate layers get deeper. However, it remains unclear whether the projector exhibits a similar trend. To quantitatively evaluate the clustering performance of these components, we employ widely recognized metrics such as the Silhouette Coefficient (SC) (Rousseeuw, 1987) and Adjusted Rand Index (ARI) (Hubert & Arabie, 1985). In particular, a larger mean value of SC ($SC_{mean}$) indicates stronger **local** clustering ability in the representation, and a smaller standard deviation ($SC_{std}$) reflects better stability in local clustering [1]. Meanwhile, higher ARI values correspond to enhanced **global** clustering properties. The detailed introduction of these metrics can be found in the Appendix A.

Using these metrics, we first evaluate the clustering abilities of *encoding* **H** and *embedding* **Z** across various self-supervised pretrained models in CIFAR-10 (Krizhevsky et al., 2009) dataset, which contains only 10 classes and is commonly used for cluster analysis (Ben-Shaul et al., 2023). It is evident in Figure 3 that, across most SSL models, *encodings* achieve visibly higher ARI and $SC_{mean}$, as well as lower $SC_{std}$ values, compared to *embeddings*. These observations reflect common grounds of SSL models: *Encodings* not only

---

[1]The 'clustering ability' refers to how well the vectors can represent the underlying structure of the data, while 'stability' signifies that the clustering results are more consistent across the data points, meaning fewer outliers and more stable cluster assignments.

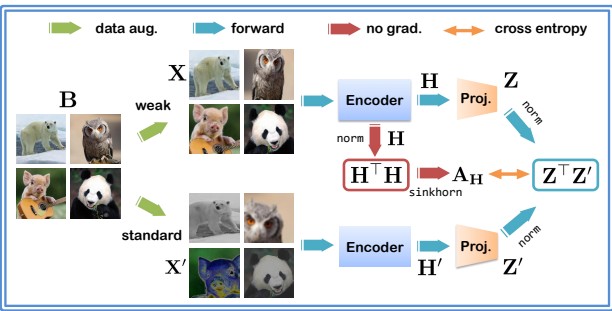

*Figure 5.* The framework of **Re**presentation **S**elf-**A**ssignment (ReSA). Here, *no grad.* denotes that the operation does not involve gradient propagation, *norm* signifies that each sample is $L_2$-normalized to compute cosine similarities, and *sinkhorn* refers to the Sinkhorn-Knopp algorithm used for clustering assignment.

possess richer semantic information but also demonstrate high-quality clustering properties. These include excellent local clustering ability ($SC_{mean}$) and stability ($SC_{std}$), global clustering capability and similarity measure effectiveness (ARI). An exception to this pattern is observed with SwAV and BYOL, whose *embeddings* also perform well across these metrics. We speculate that this may be due to the design of their loss functions, which enables the *embeddings* to learn effective clustering properties. For instance, SwAV learns by predicting cluster assignments directly on the *embeddings*.

To gain deeper insights into the evolution of clustering properties of each component during training, we conduct experiments on the CIFAR-100 (Krizhevsky et al., 2009) dataset, which features a more complex set of categories, using three methods: SimCLR, VICReg, and SwAV. The results are shown in Figure 4. Overall, the *encodings* demonstrate a consistent improvement in all clustering metrics throughout training. In contrast, the ones of *embeddings* degrade significantly during the later stages of training. Moreover, the clustering metrics of the *hidden layer outputs within the projector* show notable differences and weaknesses compared to those of the *encodings*, despite $P_0$ being separated by only a single linear layer and achieving nearly the same linear evaluation accuracy as the *encodings*.

In summary, the experiments above demonstrate that *encodings* exhibit superior and more stable clustering properties compared to *embeddings* and the *hidden layer outputs within the projector* across various SSL models. This finding highlights the potential of leveraging *encoding* as the optimal representation for clustering, providing a foundation for designing positive-feedback SSL systems that capitalize on these robust clustering properties.

## 4. Leverage Clustering Properties for Positive-Feedback Learning

Based on above analyses, we design a novel positive-feedback SSL method, which derives **Re**presentation **S**elf-

**A**ssignment (ReSA) to guide the loss function among *embeddings* $\mathbf{Z}$ and $\mathbf{Z}'$. See Figure 5 for the clear framework.

### 4.1. Online Self-Clustering

Following notations in Section 2.1, we apply the *encoding* $\mathbf{H}$ as the representation to perform clustering. Unlike previous approaches, e.g. SwAV (Caron et al., 2020) and DINOv2 (Oquab et al., 2023) employing learnable prototypes to map features into the clustering space, we treat samples in $\mathbf{H} = [\mathbf{h}_1, \ldots, \mathbf{h}_m]$ simultaneously as points to be clustered and as **anchors**. In details, we first calculate the cosine self-similarity matrix by $\mathbf{S_H} = \mathbf{H}^\top\mathbf{H}$, where samples in $\mathbf{H}$ are $L_2-$normalized as $\mathbf{h}_i/\|\mathbf{h}_i\|_2, \forall i$. Then the online clustering assignment $\mathbf{A_H}$ is computed upon $\mathbf{S_H}$ using the iterative Sinkhorn-Knopp (Cuturi, 2013) as shown in Algorithm 1.

---

**Algorithm 1** Sinkhorn-Knopp Algorithm

---

**Require:** Cosine self-similarity matrix $\mathbf{S_H} \in \mathbb{R}^{m \times m}$, regularization parameter $\epsilon > 0$, all-ones column vector $\mathbf{1}_m$, iteration count $T$, Hadamard product $\circ$

**Ensure:** Doubly stochastic matrix $\mathbf{A_H}$

 Initialize $\mathbf{Q} \leftarrow \frac{\exp(\mathbf{S_H}/\epsilon)^\top}{\sum_{i,j}\exp(\mathbf{S_H}/\epsilon)}$

 Initialize marginals: $\mathbf{c} \leftarrow \frac{1}{m}\mathbf{1}_m$

 **for** $t = 1$ to $T$ **do**

  Compute row sums: $\mathbf{u} \leftarrow \mathbf{Q}\mathbf{1}_m$

  Normalize rows: $\mathbf{Q} \leftarrow \mathbf{Q} \circ \left(\frac{\mathbf{c}}{\mathbf{u}}\right)\mathbf{1}_m^\top$

  Compute column sums: $\mathbf{v} \leftarrow \mathbf{Q}^\top\mathbf{1}_m$

  Normalize columns: $\mathbf{Q} \leftarrow \mathbf{Q} \circ \mathbf{1}_m \left(\frac{\mathbf{c}}{\mathbf{v}}\right)^\top$

 **end for**

 Normalize columns again: $\mathbf{Q} \leftarrow \mathbf{Q} \circ \mathbf{1}_m \left(\frac{1}{\mathbf{Q}^\top\mathbf{1}_m}\right)^\top$

 Return $\mathbf{A_H} \leftarrow \mathbf{Q}^\top$

---

We follow SwAV which uses only 3 iterations and sets the regularization parameter $\epsilon = 0.05$. This algorithm does **not** involve gradient propagation, enabling it to be efficiently implemented on GPUs (Caron et al., 2020). After obtaining the doubly stochastic matrix $\mathbf{A_H}$ as the assignment, it can naturally be utilized to guide relationship between the *embeddings* $\mathbf{Z}$ and $\mathbf{Z}'$. Specially, we use the cross-entropy loss to promote the learning process:

$$\ell_{\text{ReSA}} = -\frac{1}{2m}\bigg(\sum_{i,j}\mathbf{A_H} \circ \log\mathcal{D}(\mathbf{Z}^\top\mathbf{Z}') + \sum_{i,j}\mathbf{A_H}^\top \circ \log\mathcal{D}(\mathbf{Z}'^\top\mathbf{Z})\bigg) \quad (2)$$

where $\mathcal{D}(\mathbf{Z}^\top\mathbf{Z}') = \frac{\exp(\mathbf{Z}^\top\mathbf{Z}'/\tau)}{\exp(\mathbf{Z}^\top\mathbf{Z}'/\tau)\mathbf{1}_m}$ and $\mathcal{D}(\mathbf{Z}'^\top\mathbf{Z})$ are probability distributions derived through the softmax function. $\tau$ is a scalar temperature hyperparameter, $\circ$ stands for Hadamard product, and $\mathbf{1}_m$ is the all-ones column vector.

**Comparison to SwAV.** As a pioneering SSL method based on online clustering, SwAV (Caron et al., 2020) employs a 'swapped' prediction mechanism (which is also adopted by DINOv2 (Oquab et al., 2023)), where the cluster assignment of one view is predicted from the *embedding* of another view. This is achieved by minimizing the following objective:

$$\ell_{\text{SwAV}} = -\frac{1}{2m}\bigg( \sum_{i,j} \mathbf{Q}' \circ \log \mathcal{D}(\mathbf{Z}^\top \mathbf{C}) +$$
$$\sum_{i,j} \mathbf{Q} \circ \log \mathcal{D}(\mathbf{Z}'^\top \mathbf{C}) \bigg) \tag{3}$$

where $\mathbf{C} \in \mathbb{R}^{d_g \times K}$ is the prototype matrix learned by backpropagation, and $\mathbf{Q} = \text{sinkhorn}(\mathbf{Z}^\top \mathbf{C})$ is the cluster assignment using Sinkhorn-Knopp algorithm. The key differences and advantages of our ReSA compared to SwAV (and DINOv2) can be summarized as follows: (1) ReSA computes clustering assignments on *encoding* with high-quality clustering properties, whereas SwAV performs it on the less stable *embedding*. (2) SwAV requires learnable prototypes, which often necessitate complex design strategies, such as freezing prototypes during the early stages of training and use a large number of prototypes $K$ to ensure stability, whereas ReSA directly extracts clustering information from the representations. (3) SwAV executes the Sinkhorn-Knopp algorithm multiple times, corresponding to the number of global augmented views. In contrast, ReSA only requires a single execution of this algorithm regardless of the number of augmented views. This highlights the efficiency of ReSA, particularly under multi-crop scenarios.

**Comparison to InfoNCE.** As a well-known contrastive loss function, InfoNCE (Oord et al., 2018) aims to maximize the similarity between positive pairs while minimizing the similarity between the negative pairs, thereby approximating the optimization of mutual information as follows:

$$\ell_{\text{InfoNCE}} = -\frac{1}{2m}\bigg( \sum_{i,i} \log \mathcal{D}(\mathbf{Z}^\top \mathbf{Z}') + \sum_{i,i} \log \mathcal{D}(\mathbf{Z}'^\top \mathbf{Z}) \bigg) \tag{4}$$

It is evident that when $\mathbf{A_H}$ equals the identity matrix, ReSA and InfoNCE are entirely equivalent. In other words, ReSA guides the relationship among *embeddings* through assignments obtained via self-clustering, whereas InfoNCE employs the identity matrix as a hard matching target, strictly enforcing the maximization of distances between all negative pairs, which may inadvertently push samples belonging to the same category further apart during training, thereby disrupting the underlying semantic cluster structure (Wang & Liu, 2021; Huang et al., 2024).

### 4.2. A Gradient Perspective on ReSA

Many works have conducted in-depth theoretical studies on InfoNCE, such as the alignment and uniformity properties observed by Wang & Isola (2020), the hardness-aware property discovered by Wang & Liu (2021) through gradient analysis, and the probabilistic model of InfoNCE derived by Bizeul et al. (2024) from the perspective of mutual information. Since the motivation for our proposed ReSA does not have a direct connection with the evidence lower bound of mutual information, in this section, we provide an intuitive gradient analysis to further understand the optimization mechanism of ReSA.

Given the $L_2$-normalized embedding vectors $\mathbf{Z} = [\mathbf{z}_1, \ldots, \mathbf{z}_m]$ and $\mathbf{Z}' = [\mathbf{z}'_1, \ldots, \mathbf{z}'_m]$, the InfoNCE formula on $\mathbf{z}_i$ can be writen as (omitting the symmetric terms):

$$\ell_{\text{InfoNCE}}(\mathbf{z}_i) = -\log\left( \frac{\exp(s_{i,i}/\tau)}{\sum_{k=1}^m \exp(s_{i,k}/\tau)} \right) \tag{5}$$

where $s_{i,j} = \mathbf{z}_i^\top \mathbf{z}'_j, \forall i, j$. Defining the probability $P_{i,j} = \frac{\exp(s_{i,j}/\tau)}{\sum_{k=1}^m \exp(s_{i,k}/\tau)}$, the gradients of InfoNCE with respect to the positive similarity $s_{i,i}$ and the negative similarity $s_{i,j}$ ($i \neq j$) are formulated as (Wang & Liu, 2021):

$$\frac{\partial \ell_{\text{InfoNCE}}(\mathbf{z}_i)}{\partial s_{i,i}} = -\frac{1}{\tau}\sum_{k \neq i} P_{i,k}, \quad \frac{\partial \ell_{\text{InfoNCE}}(\mathbf{z}_i)}{\partial s_{i,j}} = \frac{1}{\tau}P_{i,j} \tag{6}$$

Similarly, our ReSA formula on $\mathbf{z}_i$ can be written as:

$$\ell_{\text{ReSA}}(\mathbf{z}_i) = -\sum_{j=1}^m \mathbf{A_H}^{(i,j)} \log\left( \frac{\exp(s_{i,j}/\tau)}{\sum_{k=1}^m \exp(s_{i,k}/\tau)} \right) \tag{7}$$

By contrast, the gradient of ReSA with respect to the similarity $s_{i,j}$ for any pair of samples ($\forall i, j$) takes exactly the same analytical form:

$$\frac{\partial \ell_{\text{ReSA}}(\mathbf{z}_i)}{\partial s_{i,j}} = \frac{1}{\tau}(P_{i,j} - \mathbf{A_H}^{(i,j)}) \tag{8}$$

Based on the gradient analysis above, we know that InfoNCE explicitly distinguishes between the gradient forms of positive and negative similarities. This restrictive mechanism naturally leads to harmful gradient updates for negative sample pairs within the same class. In contrast, ReSA eliminates the distinction between positive and negative samples and adapts to optimize the similarity of all sample pairs by leveraging self-clustering of the encodings, thereby addressing a key challenge in contrastive learning.

### 4.3. Impact of Image Augmentation on ReSA

Having introduced the learning process of ReSA, it is essential to consider another critical aspect of SSL: image augmentation, which has long been acknowledged as a key factor in enhancing the performance of self-supervised methods (Chen et al., 2020b; Grill et al., 2020). Standard prac-

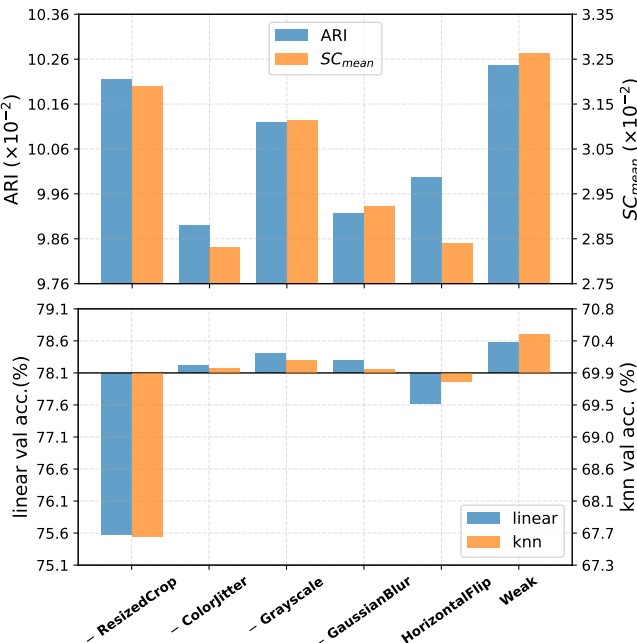

*Figure 6.* Investigate the impact of image augmentation on ReSA. The experiments are conducted employing a ResNet-18 encoder pretrained on ImageNet-100 for 200 epochs. The starting positions of bars represent results of the standard augmentation. The symbols '−' on the $x$-axis indicate the removal of a specific transformation from the standard augmentation. 'Weak' denotes the weak augmentation that includes only *ResizedCrop* and *HorizontalFlip*.

tices involve employing a variety of complex transformations with random probabilities, such as *ResizedCrop*, *ColorJitter*, *Grayscale*, *GaussianBlur*, and *HorizontalFlip*, to increase the task's complexity and improve the robustness of the learned representations. However, for clustering-based SSL methods, overly aggressive augmentations can distort the original image information, making it harder for the model to discern meaningful patterns (Zheng et al., 2021), which may result in incorrect clustering assignments. To address this, we systematically evaluate the effect of each transformation technique on ReSA's clustering performance during training and its linear evaluation accuracy.

Since ReSA only requires extracting clustering information from the *encodings* of **one single view**, we only need to adjust the image augmentation for that specific view, while the standard augmentation setting can be applied to the other view(s). Subsequently, we conduct experiments using the high-resolution ImageNet-100 (Tian et al., 2020) dataset, and the results are presented in Figure 6. It is evident that removing any single transformation improves the clustering performance of the representations, with *Resized-Crop* (replaced with fixed *CenterCrop*) having the most significant impact. However, its removal leads to a substantial decline in representation quality, indicating the critical role of *ResizedCrop* in learning invariance. The removal of *ColorJitter*, *Grayscale*, or *GaussianBlur* each results

in improvements across various metrics, whereas removing *HorizontalFlip* causes a slight drop in val accuracies. Based on these findings, we design a weak augmentation for ReSA, which includes only *ResizedCrop* and *HorizontalFlip*, and discover that this design not only significantly enhances the clustering performance of the representations but also improves their overall quality. We note that these findings align with the results observed in ReSSL (Zheng et al., 2021), where weak augmentation enables the model to better capture the relationships among samples.

In summary, we have completed the introduction of the ReSA framework as shown in Figure 5, which leverages clustering information extracted from the *encoding* to guide the design of the loss function, thereby achieving positive-feedback self-supervised learning.

## 5. Experiments on Standard SSL Benchmark

In this section, we conduct extensive experiments on standard SSL benchmarks to evaluate the effectiveness of ReSA. We perform pretraining from scratch on a variety of datasets, including CIFAR-10/100, ImageNet-100, and ImageNet (Deng et al., 2009), utilizing diverse encoders such as ConvNets and the ViT. Furthermore, we compare the performance of ReSA with state-of-the-art SSL methods across a range of downstream tasks, e.g. linear probe evaluation and transfer learning. The full PyTorch-style algorithm as well as details of implementation is provided in Appendix B.

### 5.1. Evaluation for Classification

**Evaluation on small and medium size datasets.** Following the benchmark in solo-learn (da Costa et al., 2022), we first perform pretraining and classification evaluations on CIFAR-10/CIFAR-100, and ImageNet-100, strictly adhering to the same experimental settings as other methods without introducing any additional tricks. The results in Table 1 reveal that ReSA consistently outperforms state-of-the-art methods, even those with carefully optimized parameters, across all datasets. Particularly noteworthy is ReSA's performance in *k-nearest neighbors* classification, surpassing other methods with absolute accuracy improvements of approximately $3\%$ to $8\%$ on CIFAR-10 and $5\%$ to $13\%$ on CIFAR-100. These findings highlight that ReSA captures representations with superior clustering structures.

**Evaluation on ImageNet.** Following the ImageNet evaluation protocol commonly used by SSL methods, we pretrain ResNet-50 encoders with ReSA for varying numbers of epochs. As shown in Table 2, ReSA consistently outperforms other methods on the large-scale ImageNet dataset. Remarkably, after only 100-epoch training, ReSA surpasses the performance of SimCLR, SwAV, and SimSiam trained for 800 epochs. With 200 epochs, ReSA exceeds state-of-the-art methods such as MoCoV3, Barlow Twins, VICReg,

*Table 1.* Classification top-1 accuracies of a *linear* and a *k-nearest neighbors* ($k = 5$) classifier for different loss functions and datasets. The table is mostly inherited from solo-learn (da Costa et al., 2022). All methods are based on ResNet-18 with two augmented views generated from per sample and are trained for 1000-epoch on CIFAR-10/100 with a batch size of 256 and 400-epoch on ImageNet-100 with a batch size of 128. The bold values indicate the best performance, and the underlined values represent the second highest accuracy.

| Method | CIFAR-10 | | CIFAR-100 | | ImageNet-100 | |
|---|---|---|---|---|---|---|
| | *linear* | *k-nn* | *linear* | *k-nn* | *linear* | *k-nn* |
| SimCLR (Chen et al., 2020a) | 90.74 | 85.13 | 65.78 | 53.19 | 77.64 | 65.78 |
| BYOL (Grill et al., 2020) | 92.58 | 87.40 | 70.46 | 56.46 | 80.32 | 68.94 |
| SwAV (Caron et al., 2020) | 89.17 | 84.18 | 64.88 | 53.32 | 74.28 | 63.84 |
| SimSiam (Chen & He, 2021) | 90.51 | 86.82 | 66.04 | 55.79 | 78.72 | 67.92 |
| MoCoV3 (Chen et al., 2021) | 93.10 | 89.47 | 68.83 | 58.23 | 80.36 | 72.76 |
| W-MSE (Ermolov et al., 2021) | 91.55 | 89.69 | 66.10 | 56.69 | 76.23 | 67.72 |
| DINO (Caron et al., 2021) | 89.52 | 86.13 | 66.76 | 56.24 | 74.92 | 64.30 |
| Barlow Twins (Zbontar et al., 2021) | 92.10 | 88.09 | 70.90 | 59.40 | 80.16 | 72.14 |
| VICReg (Bardes et al., 2022) | 92.07 | 87.38 | 68.54 | 56.32 | 79.40 | 71.94 |
| CW-RGP (Weng et al., 2022) | 92.03 | 89.67 | 67.78 | 58.24 | 76.96 | 68.46 |
| INTL (Weng et al., 2024) | 92.60 | 90.03 | 70.88 | 61.90 | 81.68 | 73.46 |
| **ReSA (ours)** | **93.53** | **93.02** | **72.21** | **66.83** | **82.24** | **74.56** |

*Table 2.* ImageNet classification top-1 accuracy of a *linear* classifier based on ResNet-50 encoder. All methods are pretrained with **two** $224^2$ **augmented views** generated from per sample. Given that one of the objectives of SSL methods is to achieve high performance with small batch sizes (Chen et al., 2020b; Chen & He, 2021), it's worth noting that our ReSA can also perform effectively when trained with a small batch size of 256.

| Method | batch size | pretrained epochs | | |
|---|---|---|---|---|
| | | 100 | 200 | 800 |
| SimCLR | 256 | 57.5 | 62.0 | 66.5 |
| | 4096 | 66.5 | 68.3 | 70.4 |
| SwAV | 256 | 65.5 | 67.7 | - |
| | 4096 | 66.5 | 69.1 | 71.8 |
| MoCoV3 | 1024 | 67.4 | 71.0 | 72.4 |
| | 4096 | 68.9 | - | 73.8 |
| BYOL | 4096 | 66.5 | 70.6 | 74.3 |
| Barlow Twins | 2048 | 67.7 | 70.2 | 73.2 |
| VICReg | 2048 | 68.6 | 70.8 | 73.1 |
| SimSiam | 256 | 68.1 | 70.0 | 71.3 |
| | 1024 | 68.0 | 69.9 | 71.1 |
| MEC | 256 | 70.1 | - | - |
| | 1024 | 70.6 | 71.9 | 74.0 |
| INTL | 256 | 69.5 | 71.1 | 73.1 |
| | 1024 | 69.7 | 71.2 | 73.3 |
| **ReSA (ours)** | 256 | **71.9** | **73.4** | - |
| | 1024 | **71.3** | **73.8** | **75.2** |

*Table 3.* ImageNet classification top-1 accuracy of a *linear* and a *k-nearest neighbors* ($k = 20$) classifier based on a standard ViT-S/16 encoder. All models are pretrained for 300-epoch with two $224^2$ views.

| Classifier | BYOL | SwAV | MoCoV3 | DINO | **ReSA** |
|---|---|---|---|---|---|
| *linear* | 71.4 | 68.5 | 72.5 | 72.5 | **72.7** |
| *k-nn* | 66.6 | 60.5 | 67.7 | 67.9 | **68.3** |

*Table 4.* Comparison of Computational overhead among various SSL methods. For fairness, we set the batch size to 1024 with two $224^2$ augmented views pretraining on ImageNet, and perform all measurements including peak memory (GB per GPU) and training time (hours per epoch) on the same environment and machine equipped with 8 A100-PCIE-40GB GPUs using 32 dataloading workers under mixed-precision.

| Method | encoder | memory | time |
|---|---|---|---|
| SwAV | ResNet-50 | **13.7** | 0.19 |
| | ViT-S/16 | **14.8** | 0.17 |
| MoCoV3 | ResNet-50 | 14.6 | 0.19 |
| | ViT-S/16 | 15.5 | 0.22 |
| DINO | ResNet-50 | 15.5 | 0.20 |
| | ViT-S/16 | 16.4 | 0.21 |
| **ReSA (ours)** | ResNet-50 | 14.6 | **0.16** |
| | ViT-S/16 | 15.6 | **0.12** |

### 5.2. Analysis on Computational Overhead

In Table 4, we provide a fair comparison of the training costs among ReSA and several SSL methods. The results show that ReSA has memory consumption comparable to MoCoV3 but achieves faster training speeds, especially on ViTs, where it is nearly twice as fast. This improvement is attributed to ReSA's simpler image augmentation settings and the removal of batch normalization (BN) from the projector and predictor MLPs. Additionally, ReSA outperforms DINO in both memory usage and training time. This advantage stems from DINO's reliance on an extremely high-

and INTL, all trained for 800 epochs. When extended to 800 epochs, ReSA achieves a linear classification accuracy of 75.2%, a level that methods like SwAV and DINO only reach by employing the multi-crop (Caron et al., 2020) trick. These results underscore ReSA's exceptional potential for training on large-scale datasets. Additionally in Table 3, we conduct preliminary evaluations of ReSA's training capability on the Vision Transformer, using a standard ViT-S/16, which has a comparable number of parameters to ResNet-50. We do not incorporate extensive training tricks, yet ReSA still outperform DINO in both *linear* and *k-nn* classification.

*Table 5.* Transfer Learning to COCO detection and instance segmentation. All competitive methods are based on ResNet-50 with 200-epoch pretraining on ImageNet. We follow MoCo (He et al., 2020) to apply Mask R-CNN (1 × schedule) fine-tuned in COCO 2017 train, evaluated in COCO 2017 val.

| Method | COCO detection | | | COCO instance seg. | | |
|---|---|---|---|---|---|---|
| | $AP_{50}$ | AP | $AP_{75}$ | $AP_{50}$ | AP | $AP_{75}$ |
| Scratch | 44.0 | 26.4 | 27.8 | 46.9 | 29.3 | 30.8 |
| Supervised | 58.2 | 38.2 | 41.2 | 54.7 | 33.3 | 35.2 |
| SimCLR | 57.7 | 37.9 | 40.9 | 54.6 | 33.3 | 35.3 |
| MoCoV2 | 58.8 | 39.2 | 42.5 | 55.5 | 34.3 | 36.6 |
| BYOL | 57.8 | 37.9 | 40.9 | 54.3 | 33.2 | 35.0 |
| SwAV | 57.6 | 37.6 | 40.3 | 54.2 | 33.1 | 35.1 |
| SimSiam | 59.3 | 39.2 | 42.1 | 56.0 | 34.4 | 36.7 |
| Barlow Twins | 59.0 | 39.2 | 42.5 | 56.0 | 34.3 | 36.5 |
| MEC | 59.8 | 39.8 | 43.2 | 56.3 | 34.7 | 36.8 |
| INTL | 60.9 | 40.7 | 43.7 | 57.3 | 35.4 | 37.6 |
| **ReSA (ours)** | **61.1** | **41.0** | **44.3** | **57.7** | **35.7** | **38.4** |

dimensional prototype (e.g., output dimension = 65536), which significantly impacts training efficiency. Finally, while SwAV exhibits the smallest memory footprint among the methods due to its absence of momentum networks, its training speed remains slower than ReSA. This is because SwAV also requires a higher-dimensional prototype and performs the Sinkhorn-Knopp algorithm twice per iteration.

### 5.3. Transfer to Downstream Tasks

To evaluate the quality of representations learned by ReSA, we transfer our pretrained model to downstream tasks, including COCO (Lin et al., 2014) object detection and instance segmentation. For these tasks, we adopt the baseline codebase from MoCo (He et al., 2020). Most results reported in Table 5 are inherited from SimSiam paper (Chen & He, 2021). Notably, ReSA also achieves better performance compared to other methods on both tasks, highlighting its strong potential for downstream applications.

## 6. How ReSA Shapes Better Clustering Properties?

In this section, we utilize visualizations and additional experiments to illustrate the differences among the representations learned by ReSA and other SSL methods, as well as to investigate whether and how ReSA facilitates better clustering properties.

Firstly, we track the evolution of clustering metrics for each component during ReSA training, as shown in Figure 9. It can be clearly observed that while ReSA exhibits relatively slow performance improvement in the early stages of training, it significantly outperforms other methods in the later stages. Interestingly, all components of ReSA demonstrate strong clustering properties, suggesting that leveraging high-quality clustering information from the *encodings* to guide the learning of *embeddings* enables the projector layers to

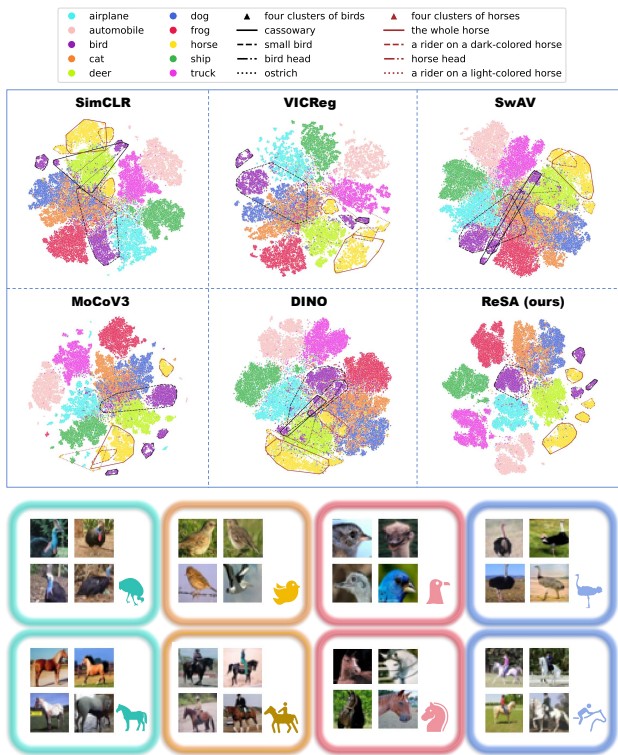

*Figure 7.* T-SNE visualization of SSL representations on CIFAR-10. All methods are pretrained for 1000 epochs on CIFAR-10 using ResNet-18, with *encodings* utilized as representations to visualize all training data. For the multiple centroids observed in the bird and horse categories, we enclose points of each subclass with convex polygons and display the corresponding images.

also acquire robust clustering performance.

### 6.1. ReSA Excels at Fine-grained Learning

Furthermore, in Figure 7, we visualize the representation distributions learned by ReSA and other SSL methods on CIFAR-10 using T-SNE (van der Maaten & Hinton, 2008). Notably, the representations learned by ReSA exhibit clear separations between different classes, whereas those learned by other methods show varying degrees of overlap, making it difficult to discern distinct boundaries.

Another intriguing observation is the presence of multiple centroids within the bird and horse categories in the representations. Upon further investigation of the samples corresponding to these centroids, we find that, unlike the neural collapse (Papyan et al., 2020) in supervised learning (where samples of the same class collapse to a single point), SSL models are capable of capturing more fine-grained features, such as color distinctions (e.g., cassowary vs. ostrich), structural differences (e.g., whole horse vs. horse head), and the presence of multiple objects (e.g., a rider on a horse). Moreover, compared to other methods, ReSA demonstrates a superior ability to distinguish these fine-grained features.

*Table 6.* Transfer learning to fine-grained datasets based on ResNet-50 pretrained on ImageNet. We employ a *k-nearest neighbors* classifier ($k = 5, 10, 20$), without requiring additional training or parameter tuning. The model weights for all other methods are sourced directly from their respective official codebases. † indicates that these methods employ the multi-crop trick, i.e. generating two $224^2$ views and six $96^2$ views for each image, which can enhance performance but comes at the cost of additional computational overhead.

| Method | pretrained epochs | ImageNet-1K | | | CUB-200-2011 | | | Pets-37 | | | Food-101 | | | Flowers-102 | | |
|---|---|---|---|---|---|---|---|---|---|---|---|---|---|---|---|---|
| | | 5 | 10 | 20 | 5 | 10 | 20 | 5 | 10 | 20 | 5 | 10 | 20 | 5 | 10 | 20 |
| MoCoV3 | 1000 | 67.9 | 68.9 | 68.9 | 46.8 | 48.8 | 50.4 | 85.4 | 86.5 | 86.5 | 56.3 | 58.6 | 59.7 | 83.4 | 81.6 | 80.9 |
| VICReg | 1000 | 64.3 | 65.2 | 65.6 | 33.4 | 35.4 | 36.3 | 81.5 | 82.0 | 82.3 | 56.9 | 59.6 | 61.0 | 83.4 | 83.2 | 82.6 |
| INTL | 800 | 63.6 | 64.8 | 65.1 | 26.7 | 28.0 | 29.4 | 78.4 | 79.5 | 79.7 | 55.6 | 58.1 | 59.2 | 78.8 | 77.6 | 77.2 |
| **ReSA (ours)** | 800 | **69.2** | **69.9** | **69.9** | **56.5** | **58.5** | **59.9** | **85.8** | **87.2** | **87.5** | **58.3** | **60.4** | **61.3** | **84.4** | **83.6** | **83.6** |
| SwAV† | 800 | 64.3 | 65.5 | 65.7 | 26.2 | 27.3 | 28.4 | 77.2 | 77.3 | 77.1 | 54.7 | 57.4 | 58.7 | 79.3 | 79.9 | 78.4 |
| DINO† | 800 | 66.4 | 67.4 | 67.6 | 33.8 | 35.5 | 36.8 | 81.1 | 81.6 | 80.9 | 58.2 | **60.8** | **61.8** | **84.8** | **84.1** | **83.7** |

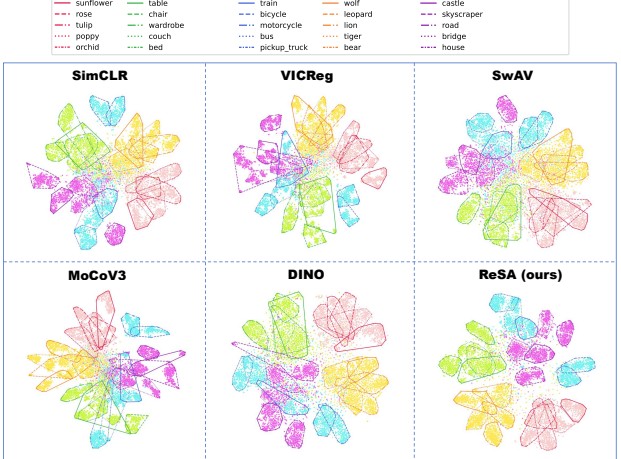

*Figure 8.* T-SNE visualization of SSL representations on CIFAR-100. We enclose points of each subclass with convex polygons.

To further substantiate this, we transfer models pretrained on ImageNet to fine-grained datasets for evaluation. As shown in Table 6, ReSA consistently outperforms other SSL methods on fine-grained datasets, with particularly considerable improvements observed on the CUB-200-2011.

### 6.2. ReSA also Stands Out in Coarse-grained Representations

Finally, we explore ReSA's performance at the coarse-grained level. Specifically, we select CIFAR-100 for visualization, as its 100 classes can be grouped into 20 coarse-grained superclasses. For clarity, we randomly selected 5 superclasses for T-SNE visualization, as shown in Figure 8. It is evident that all other methods exhibit dense overlap on the CIFAR-100 dataset, resulting in numerous indistinguishable outliers. In contrast, our ReSA effectively identifies these hard samples, clustering them correctly within their respective groups. We believe this capability of ReSA is a key factor behind its superior accuracy with the *k-nn* classifier, substantially exceeding other SSL methods. Furthermore, as shown in Table 7, we evaluate the performance of various SSL models using coarse-grained labels and observe

*Table 7.* CIFAR-100 classification top-1 accuracy of a *linear* and a *k-nearest neighbors* ($k = 5$) classifier based on 100 fine-grained classes and 20 coarse-grained superclasses.

| Method | fine-grained | | coarse-grained | |
|---|---|---|---|---|
| | *linear* | *k-nn* | *linear* | *k-nn* |
| SimCLR | 65.8 | 53.2 | 72.5 | 67.2 |
| SwAV | 64.9 | 53.3 | 70.0 | 66.3 |
| MoCoV3 | 68.8 | 58.2 | 76.4 | 68.6 |
| DINO | 66.8 | 56.2 | 72.9 | 70.2 |
| VICReg | 68.5 | 56.3 | 74.3 | 69.9 |
| **ReSA (ours)** | **72.2** | **66.8** | **79.8** | **78.8** |

that ReSA consistently achieves much higher accuracies than its counterparts. These experimental results confirm that ReSA also demonstrates exceptional performance in coarse-grained learning.

We also present ablation studies, along with a discussion of potential future research presented in Appendix C.

## 7. Conclusion

In this work, we demonstrate the feasibility of leveraging the rich clustering properties inherent in SSL models, particularly within *encodings*, to enable a positive-feedback mechanism. Building upon this, we propose ReSA, which exhibits exceptional performance across a wide range of benchmarks and excels at both fine-grained and coarse-grained learning. We believe this dual capability would take a step toward addressing the long-standing challenge of reconciling the seemingly conflicting demands of fine-grained and coarse-grained visual representations within a unified framework, thereby advancing the development of large-scale visual foundation models.

## Acknowledgment

This work was supported by the National Science and Technology Major Project (2022ZD0116306), National Natural

Science Foundation of China (Grant No. 62476016 and 62441617), the Fundamental Research Funds for the Central Universities.

## Impact Statement

This paper presents work aimed at advancing the field of self-supervised learning. By improving representation learning, our methods can contribute to a wide range of applications across various domains, including computer vision, natural language processing, and beyond. While our work has potential societal implications, such as enabling more efficient use of data and reducing reliance on labeled datasets, we do not identify any specific ethical concerns or negative consequences that require particular attention at this stage.

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

## A. Details of Clustering Metrics

To quantitatively compare the clustering ability of the *encoding*, *embedding*, and the *hidden layer outputs* within the projector, we first define the following metrics in supervised settings (assign samples with the same true label to the same cluster). Let $\mathcal{X} = \{x_1, x_2, \ldots, x_N\}$ be a set of $N$ data points, and $\mathcal{Y} = \{y_1, y_2, \ldots, y_N\}$ be the corresponding set of true labels.

**Definition A.1.** (**Silhouette Coefficient, SC**) The Silhouette Coefficient (Rousseeuw, 1987) is a measure of how similar a sample is to its own cluster compared to its nearest cluster. For a given data point $x_i$, $sc(x_i)$ is defined as:

$$sc(x_i) = \frac{b(x_i) - a(x_i)}{\max(a(x_i), b(x_i))} \tag{9}$$

where $a(x_i)$ is the average distance from point $x_i$ to all other points $x_j$ that share the same true label $y_i = y_j$, and $b(x_i)$ is the minimum average distance from point $x_i$ to all points $x_j$ that have a different true label $y_i \neq y_j$.

Based on this definition, we know that SC focuses on measuring the local clustering ability of features, with higher values indicating better **local** clustering ability. Although the SC ranges from $[-1, 1]$, the diversity of features learned through SSL can cause a large value of $\max(a(x_i), b(x_i))$, leading to a smaller effective range for SC. Therefore, in this paper, we note that SC $> 0$ indicates that the sample has been assigned to the correct cluster. For population statistics, we can compute the mean $\text{SC}_{\text{mean}}$ and standard deviation $\text{SC}_{\text{std}}$ over all data points in $\mathcal{X}$.

**Definition A.2.** (**Adjusted Rand Index, ARI**) The Adjusted Rand Index (Hubert & Arabie, 1985) is a measure of the agreement between two partitions of data, adjusted for chance grouping. Given the true labels $\mathcal{Y}$ and a set of predicted labels $\mathcal{Y}' = \{y_1', y_2', \ldots, y_N'\}$, the ARI is defined as:

$$\text{ARI} = \frac{\text{RI} - \mathbb{E}[\text{RI}]}{\max(\text{RI}) - \mathbb{E}[\text{RI}]} \tag{10}$$

where RI is the Rand Index, and $\mathbb{E}[\text{RI}]$ is its expected value under random labeling.

In practice, the ARI can be computed using a contingency table between $\mathcal{Y}$ and $\mathcal{Y}'$. Let $n_{ij}$ denote the number of data points assigned to the $i$-th cluster in $\mathcal{Y}$ and the $j$-th cluster in $\mathcal{Y}'$. Defining $a_i = \sum_j n_{ij}$ and $b_j = \sum_i n_{ij}$, then the ARI is calculated as:

$$\text{ARI} = \frac{\sum_{i,j} \binom{n_{ij}}{2} - \left( \sum_i \binom{a_i}{2} \sum_j \binom{b_j}{2} \Big/ \binom{N}{2} \right)}{\frac{1}{2} \left( \sum_i \binom{a_i}{2} + \sum_j \binom{b_j}{2} \right) - \left( \sum_i \binom{a_i}{2} \sum_j \binom{b_j}{2} \Big/ \binom{N}{2} \right)}$$

where $\binom{n}{2} = \frac{n(n-1)}{2}$ is the binomial coefficient. The ARI ranges from $[-0.5, 1]$, where an ARI close to 1 indicates a perfect agreement between the true and predicted labels, and an ARI close to 0 suggests that the prediction is no better than random assignment.

Typically, we apply $k$-means ($k$ is set to the true number of classes) clustering on $\mathcal{X}$ to obtain the pseudo labels $\mathcal{Y}'$. Subsequently, ARI is used to measure the agreement between true and pseudo labels, thereby reflecting **global** clustering ability of the features and the extent to which **similarity measures** effectively capture the data structure.

## B. Details of Implementation

In this section, we provide the details and hyperparameters for ReSA pretraining and downstream evaluation.

### B.1. Datasets

- CIFAR-10 and CIFAR-100 (Krizhevsky et al., 2009), two small-scale datasets composed of $32 \times 32$ images with 10 and 100 classes, respectively.

- ImageNet-100 (Tian et al., 2020), a random 100-class subset of ImageNet (Deng et al., 2009).

- ImageNet (Deng et al., 2009), the well-known largescale dataset with about 1.3M training images and 50K test images, spanning over 1000 classes.

- COCO2017 (Lin et al., 2014), a large-scale object detection, segmentation, and captioning dataset with 330K images containing 1.5 million object instances.

- We also evaluate on fine-grained datasets including CUB-200-2011 (Wah et al., 2011), Oxford-IIIT-Pets (Parkhi et al., 2012), Food-101 (Bossard et al., 2014), and Oxford-Flowers (Nilsback & Zisserman, 2008).

### B.2. Implementation Details of ReSA Pretraining

For clarity, we first provide the algorithm of ReSA in PyTorch-style pseudo-code:

```python
# E, Em: encoder, momentum encoder
# G, Gm: projector mlp, momentum projector mlp
# P: predictor mlp (optional)
# Tw, T: weak, standard augmentation
# temp: temperature = 0.4

for x in loader:  # load a minibatch x with m samples
    x1, x2 = Tw(x), T(x)  # two augmentation views
    h1, h2 = E(x1), E(x2)  # encodings
    z1, z2 = G(h1), G(h2)  # embeddings (mxd)
    if P: z1, z2 = P(z1), P(z2)  # predicted embeddings

    with torch.no_grad():
        update_momentum_params(0.996 -> 1)  # exponential moving average
        h1m, h2m = Em(x1), Em(x2)  # momentum encodings
        z1m, z2m = Gm(h1m), Gm(h2m)  # momentum embeddings (mxd)
        assign = sinkhorn(cos_sim(h1, h1m))  # compute representation assignment

    loss = cross_entropy(cos_sim(z1, z2m) / temp, assign) + \
           cross_entropy(cos_sim(z2, z1m) / temp, assign)
    return loss / 2

def cos_sim(x, y):
    return norm(x) @ norm(y).T  # L2-normalize

def cross_entropy(x, y):
    loss = mean(y * log_softmax(x, dim=1) + y.T * log_softmax(x.T, dim=1))
    return - loss / 2

def sinkhorn(scores, eps=0.05, niters=3): # Here 'scores' should be a square matrix
    Q = exp(scores / eps).T
    Q /= Q.sum()
    m , _ = Q.shape
    c = ones(m) / m
    for _ in range(niters):
        u = Q.sum(dim=1)
        Q *= (c / u).unsqueeze(1)
        Q *= (c / Q.sum(dim=0)).unsqueeze(0)
    return (Q / Q.sum(dim=0, keepdim=True)).T
```

**Universal settings.** In all experiments conducted in Section 5, we adopt a momentum network, consistent with the practices of most existing self-supervised learning (SSL) methods (Grill et al., 2020; Chen et al., 2021; Caron et al., 2021; Liu et al., 2022; Weng et al., 2024). While the momentum network is not necessary to prevent collapse in ReSA, it has been shown to effectively promote long-term learning in SSL models (He et al., 2020; Chen & He, 2021). The momentum coefficient, temperature, and Sinkhorn-Knopp parameters in ReSA are configured in accordance with the pseudo-code provided earlier, without requiring further tuning. Furthermore, a standard three-layer MLP is employed as the projector, featuring a hidden layer dimension of 2048 and an output embedding dimension of 512. For training on large-scale datasets, such as ImageNet-1K, we follow the practices of MoCoV3 (Chen et al., 2021) and MEC (Liu et al., 2022), appending a two-layer MLP predictor to the projector in ReSA. The hidden layer and embedding dimensions of the predictor are kept identical to those of the projector. Additionally, we adopt a conventional configuration: when using ConvNets as the encoder, batch normalization (BN) and ReLU activation are incorporated into the hidden layers of both the projector and predictor. However, for ViT-based encoders, we draw inspiration from DINO (Caron et al., 2021), omitting BN and replacing ReLU with the gaussian error linear units (GELU) activation. This modification ensures that ReSA operates as a *BN-free* system during ViT training, eliminating the need for BN synchronization and offering an improvement in training efficiency.

Building on the aforementioned settings, the only modifications required pertain to optimizer-related parameters, including the learning rate, weight decay, and the number of warmup epochs. These adjustments are made in accordance with the specific encoder architecture and dataset used. Nonetheless, certain settings remain fixed. For instance, we adopt the linear scaling rule, setting the learning rate as $lr = base\ lr \times batch\ size/256$. After the warmup phase, the learning rate follows a cosine decay schedule (Loshchilov & Hutter, 2017).

**Details in training ConvNets.** We follow the exact same optimization settings and parameters as INTL (Weng et al., 2024) when pretraining ConvNets. The SGD optimizer is used and detailed parameters are provided in Table 8. The only exception is when training ResNet-50 on ImageNet for 800 epochs, where we reduce the *base lr* to $0.4$ to ensure training stability. Additionally, it is worth noting that we observe a slight performance drop in ReSA when the learning rate decreases to a very small value during the later stages of training. We hypothesize that an excessively small learning rate may amplify the regularization effect of weight decay in SGD, causing the weights to diverge from the optimal solution. To address this, we set the minimum learning rate in the cosine decay schedule to $0.1 * lr$.

*Table 8.* Optimizer-related parameters in ReSA pretraining.

| Method | dataset | encoder | predictor | optimizer | *batch size* | *base lr* | weight decay | warmup |
|--------|---------|---------|-----------|-----------|------------|-----------|--------------|--------|
| ReSA | CIFAR-10 | ResNet-18 | ✗ | SGD | 256 | 0.3 | $10^{-4}$ | 2 epochs |
| | CIFAR-100 | ResNet-18 | ✗ | SGD | 256 | 0.3 | $10^{-4}$ | 2 epochs |
| | ImageNet-100 | ResNet-18 | ✗ | SGD | 128 | 0.5 | $2.5 \times 10^{-5}$ | 2 epochs |
| | ImageNet | ResNet-50 | ✓ | SGD | 256 | 0.5 | $10^{-5}$ | 2 epochs |
| | | | | | 1024 | 0.5 | $10^{-5}$ | 10 epochs |
| | | ViT-S/16 | ✓ | AdamW | 1024 | $5 \times 10^{-4}$ | 0.1 | 40 epochs |

**Details in training ViTs.** Vision Transformer (ViT) pre-training involves numerous intricate settings, such as initialization methods and optimization parameters, which have a significant impact on training outcomes. Notably, representative SSL methods on ViTs, such as DINO and MoCoV3, adopt totally distinct designs in both architecture and training strategies. DINO (Caron et al., 2021) leverages a range of training tricks, including weight decay scheduling, gradient clipping, and stochastic depth, among others. To stabilize training, it avoids mixed-precision training, which substantially reduces computational efficiency and increases memory requirements. MoCoV3 (Chen et al., 2021), on the other hand, proposes freezing the patch embedding layer to enhance training stability while enabling mixed-precision training. However, it still incorporates batch normalization (BN) in the projector and predictor MLPs, which reduces training efficiency and complicates its application in multi-view scenarios (Morningstar et al., 2024). Taking these considerations into account, we adopt the ViT design and initialization approach of MoCoV3 for ReSA, but remove the BN layers from the MLPs. This ensures that ReSA functions as a *BN-free* system during ViT training, eliminating the need for BN synchronization and improving overall training efficiency. In this paper, we set the optimizer-related parameters as shown in Table 8. We

believe that further exploration of more suitable initialization methods and related parameters for ReSA could enhance its performance, as evidenced by its outstanding results on ConvNets.

### B.3. Implementation Details of ReSA Evaluating

**Details in evaluating CIFAR-10/100.** When evaluating on CIFAR-10/100, we adopt the same linear evaluation protocol as in W-MSE (Ermolov et al., 2021) and INTL (Weng et al., 2024): training a linear classifier for 500 epochs on each labeled dataset using the Adam optimizer, without data augmentation. The learning rate is exponentially decayed from $10^{-2}$ to $10^{-6}$ and the weight decay is $5 \times 10^{-6}$. Under these settings, a single-GPU evaluation takes under one minute—substantially faster than the protocol in solo-learn (da Costa et al., 2022), which can take tens of minutes. We also apply this evaluation protocol to models provided by solo-learn; however, their performance degrades noticeably, so we report the official results in Table 1. In addition, following W-MSE and INTL, we evaluate a simple *5-nn* classifier ($k = 5$) on these datasets for completeness. We track both *linear* and *k-nn* classifier accuracies for ReSA throughout training and observe that at certain checkpoints, ReSA achieves even higher performance than the final values reported in Table 1 (e.g., linear accuracies of 93.89% on CIFAR-10 and 72.5% on CIFAR-100). Nevertheless, we report only the final checkpoint's results for consistency.

*Table 9.* Optimal learning rate (*lr*) for training linear classifiers.

| Method | pretrained settings | | | linear eval. |
| | dataset | encoder | *batch size* | *lr* |
|---|---|---|---|---|
| | ImageNet-100 | ResNet-18 | 128 | 5 |
| ReSA | | ResNet-50 | 256 | 10 |
| | ImageNet | | 1024 | 40 |
| | | ViT-S/16 | 1024 | 0.03 |

**Details in evaluating ImageNet-100/1K.** For linear evaluation, we train the linear classifier for 100 epochs with SGD optimizer and using *MultiStepLR* scheduler with $\gamma = 0.1$ dropping at the last 40 and 20 epochs. In all our linear classifier training, we fix the batch size at 256 and set the weight decay to 0. However, when using different pretraining datasets, encoders, or batch sizes, the optimal learning rate for evaluation varies accordingly. The specific optimal values for each setting are provided in Table 9. In addition, when training the linear classifier with ViT-S/16, we follow BERT (Devlin, 2018) and DINO (Caron et al., 2021) by concatenating the [CLS] tokens from the last four layers.

*Table 10.* Low-shot evaluation. All models are pretrained on ImageNet with ResNet-50, and then fine-tuned with a linear classifer on 1% or 10% subset of ImageNet for 20 epochs. † indicates employing the multi-crop trick during pretraining.

| Method | top-1 | | top-5 | |
| | 1% | 10% | 1% | 10% |
|---|---|---|---|---|
| SimCLR | 48.3 | 65.6 | 75.5 | 87.8 |
| BYOL | 53.2 | 68.8 | 78.4 | 89.0 |
| SwAV† | 53.9 | 70.2 | 78.5 | 89.9 |
| Barlow Twins | 55.0 | 69.7 | 79.2 | 89.3 |
| VICReg | 54.8 | 69.5 | 79.4 | 89.5 |
| INTL | 55.0 | 69.4 | 80.8 | 89.8 |
| **ReSA (ours)** | **56.4** | **70.4** | **81.0** | **90.1** |

We further evaluate the low-shot learning capability of ReSA in semi-supervised classification. Specifically, we fine-tune the pre-trained ReSA encoder and train a linear classifier for 20 epochs, using 1% and 10% subsets of ImageNet, following the same splits as SimCLR (Chen et al., 2020b). The optimization is conducted using the SGD optimizer with a learning rate of 0.0002 for the encoder and 40 for the classifier, under a batch size of 256, along with a cosine decay schedule. The results, presented in Table 10, demonstrate that ReSA also performs effectively in low-shot learning scenarios.

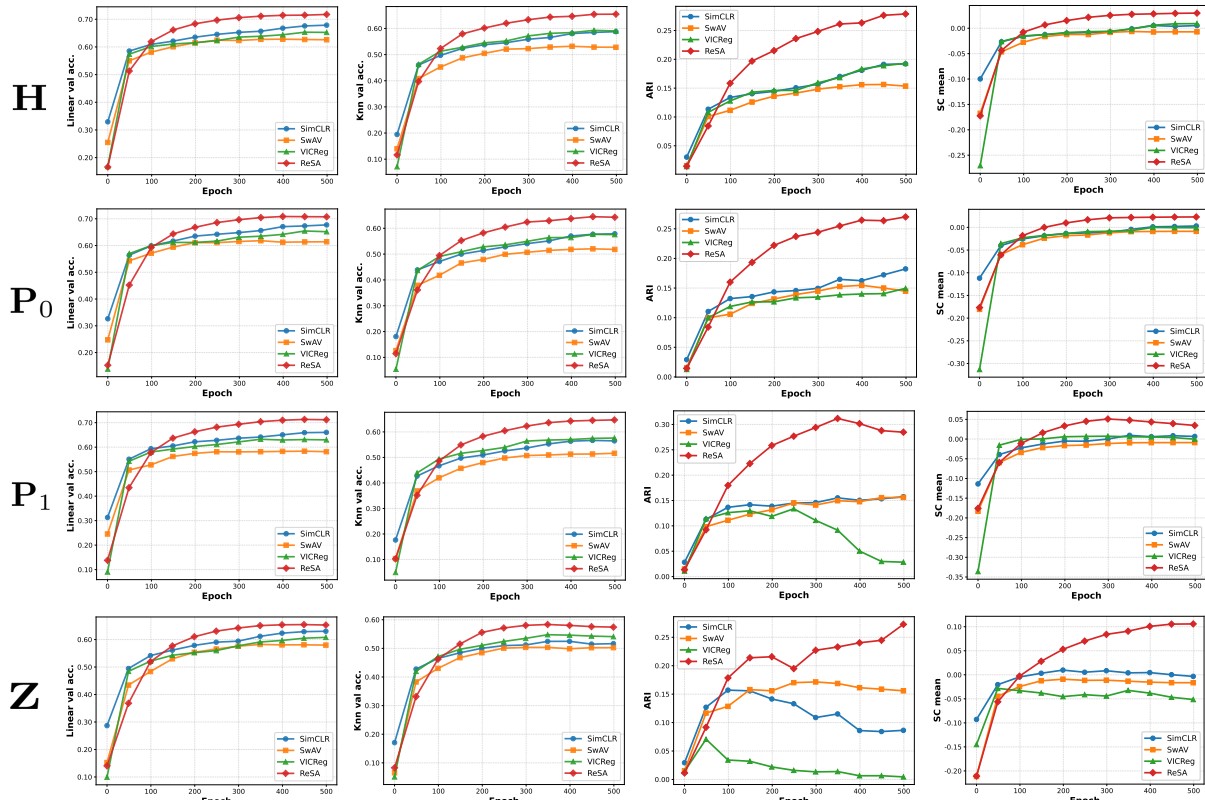

*Figure 9.* Comparison of evaluation accuracies and clustering metrics among various SSL methods during the training process. The experiments are conducted using SimCLR, SwAV, VICReg, and ReSA. The settings and notations are consistent with ones in Figure 4.

**Details in evaluating fine-grained datasets.** In the evaluation experiments on fine-grained datasets presented in Table 6, we apply a weighted *k-nearest neighbors* classifier following (Wu et al., 2018). We freeze the pretrained model to compute and store the features of the training data and use these features to select the nearest neighbors for the data in the test set. Based on the top $k$-nearest neighbors ($\mathcal{N}_k$), predictions are made using a weighted voting mechanism. Specifically, the class $c$ receives a total weight of $\sum_{i \in \mathcal{N}_k} \alpha_i \mathbf{1}_{c_i=c}$, where $\alpha_i$ represents the contribution weight. We compute $\alpha_i = \exp\left(\frac{T_i \cdot x}{\tau}\right)$, with $\tau$ set to 0.07 as described in (Wu et al., 2018) and used by DINO, without tuning this value.

## C. Additional analyses on ReSA

### C.1. Ablation Studies

As presented in Table 11, we conduct ablation studies to investigate the impact of network architecture design and temperature hyperparameter selection on ReSA performance. Our analysis reveals that employing a momentum network yields an accuracy improvement of approximately 2%, albeit at the cost of increased computational overhead. In contrast, the integration of an additional predictor demonstrates a comparable accuracy gain of around 1% while maintaining near-identical computational efficiency, exhibiting negligible impact on runtime performance.

Meanwhile, in contrast to contrastive learning methods and other approaches such as SwAV and DINO, which typically require a small temperature value (e.g. $\tau = 0.1$), ReSA achieves favorable performance with a higher temperature value ($\tau = 0.4$). This indicates that the optimization process of ReSA can effectively incorporate a broader range of samples with better tolerance, rather than focusing exclusively on hard samples (Wang & Liu, 2021), as is the case with other methods.

Additionally, we conduct experiments to test the extraction of clustering information from *embedding* to obtain the self assignment $\mathbf{A_H}$. The results shown in Figure 10 indicate that, under this condition, the loss struggles to converge, and the model's accuracy significantly declines compared to ReSA. This finding is consistent with the analyses in Section 3, where we note that the clustering properties of *embedding* are less stable and inferior to those of *encoding*, making it challenging for the model to effectively learn high-quality clustering information.

*Table 11.* Ablation studies on the network architecture and temperature hyperparameter of ReSA on ImageNet using ResNet-50 as the encoder. When evaluating the impact of different network architectures on ReSA, we set the batch size to 256 and perform pretraining using a single GPU.

| Method | momentum | predictor | *linear acc.* | memory (GB) | time (h/epoch) |
|--------|----------|-----------|---------------|-------------|----------------|
| | ✓ | ✓ | **71.9** | 25.2 | 0.78 |
| ReSA | ✓ | ✗ | 70.8 | 25.2 | 0.78 |
| | ✗ | ✓ | 69.7 | 24.3 | 0.58 |
| | ✗ | ✗ | 68.7 | 24.3 | 0.58 |

| Method | batch size | temperature $\tau$ | | | |
|--------|------------|-----|-----|-----|-----|
| | | 0.2 | 0.3 | 0.4 | 0.5 |
| ReSA | 256 | 71.2 | 71.4 | **71.9** | 71.7 |
| | 1024 | 70.9 | 71.1 | **71.3** | 71.2 |

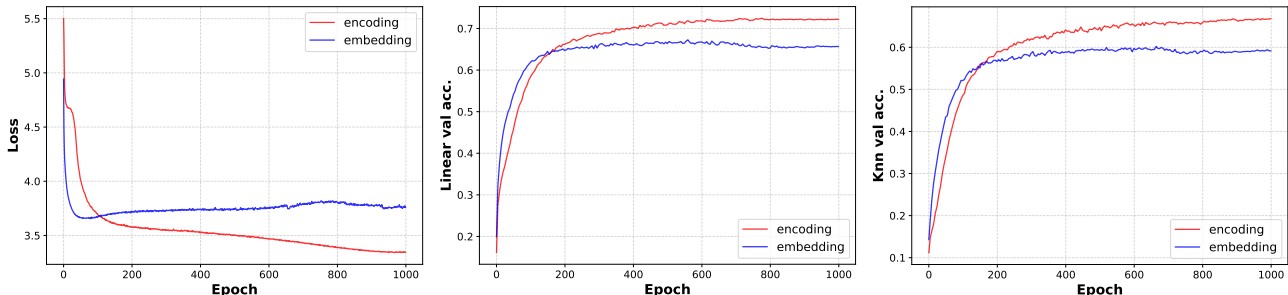

*Figure 10.* Ablation on extraction of clustering information from *encoding* vs. *embedding* to obtain the self-assignment $\mathbf{A_H}$. Both are pretrained for 1000 epochs on CIFAR-100 using ResNet-18 under totally the same experimental settings provided in Appendix B.

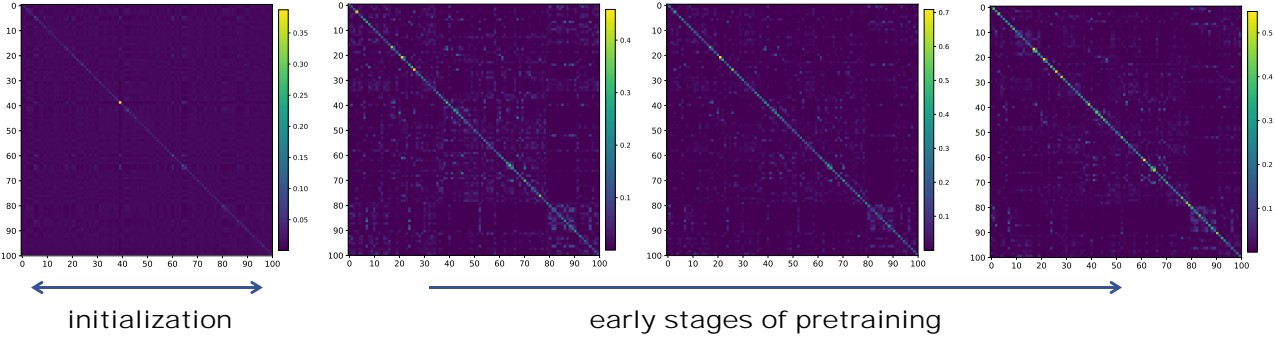

*Figure 11.* Visualization of the self-assignment matrix $\mathbf{A_H}$ during the early stages of training.

## C.2. How ReSA Avoids Feature Collapse and Early Clustering Error?

Although ReSA has demonstrated good performance in various experiments, it remains unclear how the model avoids feature collapse and clustering error in the early stages of training when it has not yet learned any clustering information.

To answer these questions, we first consider the loss formula of ReSA: $\ell_{\text{ReSA}}(\mathbf{z}_i) = - \sum_{j=1}^{m} \mathbf{A_H}^{(i,j)} \log \left( \frac{\exp(s_{i,j}/\tau)}{\sum_{k=1}^{m} \exp(s_{i,k}/\tau)} \right)$.

Here, $\mathbf{A_H}^{(i,j)}$ denotes the $(i,j)$-th element of $\mathbf{A_H}$, where $\mathbf{A_H} = \text{Sinkhorn}(\mathbf{H}^{\top}\mathbf{H})$. Given that the Sinkhorn-Knopp algorithm exhibits strict monotonicity, we have the property that if $\mathbf{h}_i^{\top}\mathbf{h}_j > \mathbf{h}_i^{\top}\mathbf{h}_k$, then $\mathbf{A_H}^{(i,j)} > \mathbf{A_H}^{(i,k)}$. Since the vectors in $\mathbf{H}$ are $L_2$-normalized, the diagonal elements of $\mathbf{H}^{\top}\mathbf{H}$ are all maximized to 1. This implies that for any $i$ and $j$, we have $\mathbf{A_H}^{(i,i)} \geq \mathbf{A_H}^{(i,j)}$. Furthermore, due to the sharp distribution employed in Sinkhorn, the diagonal elements are significantly larger than the off-diagonal elements. This ensures that the optimization focus of ReSA remains on the alignment of augmented views from the same image, substantially reducing the impact of early assignment errors (off-diagonal elements) on the training process. Throughout training, the continual alignment enables the model to learn meaningful representations, which in turn facilitates correct cluster assignments in the later stages, further promoting learning. In contrast, other clustering-based methods such as SwAV and DINO require an initialized prototype for cluster assignment, making it more challenging to avoid early clustering errors. We speculate that this is one of the key reasons why ReSA outperforms these methods in terms of overall effectiveness.

We then visualize the self-assignment matrix $\mathbf{A_H}$ during the early stages of training. As shown in Figure 11, at model initialization, the assignment already exhibits a dominance of diagonal elements, reflecting an optimization trend that pulls augmented views from the same image closer together. As training progresses, we observe that the dominance of diagonal elements gradually strengthens, effectively preventing feature collapse in the model.

### C.3. Pretraining on Long-tailed Dataset

To further evaluate the training performance of ReSA on imbalanced datasets, we conduct experiments using four self-supervised learning methods—ReSA, MoCoV3, INTL, and VICReg—pretrained and evaluated on the long-tailed CIFAR100-LT dataset. We follow the setup on `https://huggingface.co/datasets/tomas-gajarsky/cifar100-lt`, setting an imbalance factor of 1/20, resulting in the CIFAR100-LT dataset containing 15,907 images. We train these four SSL models for 1000 epochs with ResNet-18 as the encoder and evaluate it on the full CIFAR100 test set. As observed in Figure 12, the loss values of these four methods converge well, but the final evaluation accuracy is significantly lower compared to training on the full CIFAR100 dataset in Table 1. Interestingly, we notice that ReSA's loss decreases more slowly in the early stages, and its accuracy improves more gradually than other methods. We hypothesize that this may be due to noisy initial clusters in the early stages of training, causing clustering errors. However, we find that as training progresses, ReSA's accuracy continues to rise in the mid-phase, surpassing all other methods. This suggests that ReSA is able to gradually resolve these issues and learn the correct clustering patterns as training advances, rather than amplifying errors. Overall, this experiment demonstrates that ReSA can also learn effective representations on long-tailed datasets.

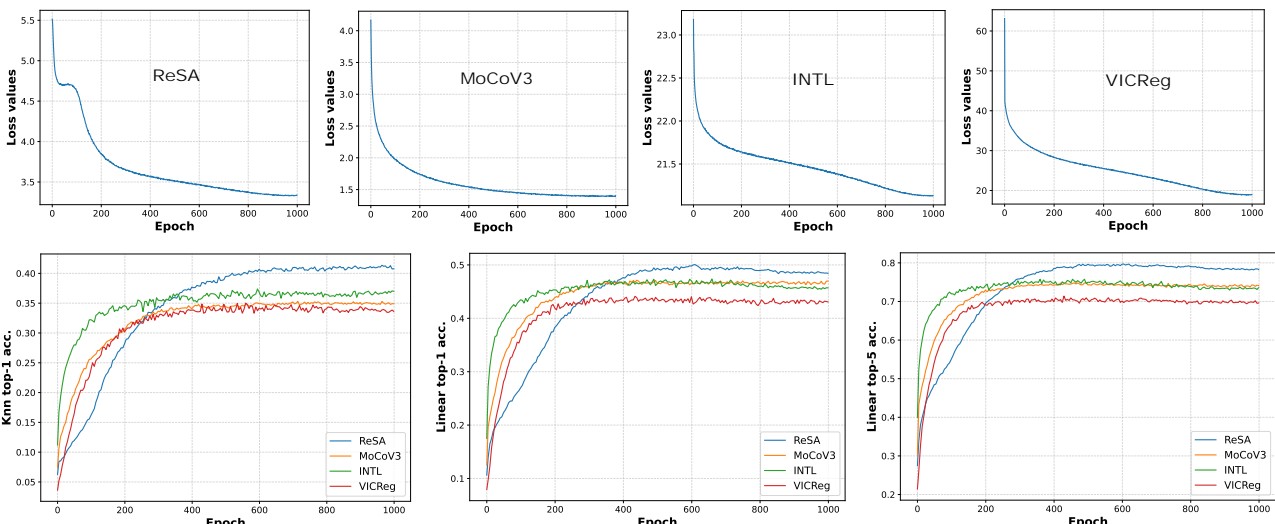

*Figure 12.* Pretraining on the long-tailed dataset. Here We report the training losses of four self-supervised learning methods on the CIFAR100-LT dataset, along with their evalutaion performance on the full CIFAR-100 test set as measured by a linear probe and a k-NN classifier.

## C.4. The impact of weak augmentation on other methods

Since the design of weak augmentation (identical to that in ReSSL (Zheng et al., 2021)) provides a certain degree of performance improvement for ReSA, we also examine its impact on other self-supervised learning methods. Specifically, we select SwAV, VICReg, and MoCoV3, strictly following the experimental configurations in solo-learn (da Costa et al., 2022), with the only modification being the replacement of the image augmentation settings. As shown in Table 12, weak augmentation do not enhance the performance of these methods. This is likely because the standard augmentation settings have been extensively tuned through numerous experiments, ensuring optimal training conditions for these approaches.

Table 12. Investigate the impact of weak augmentation on other methods.

| Method | weak aug. | CIFAR-10 | | CIFAR-100 | |
|--------|-----------|----------|------|-----------|------|
| | | *linear* | *k-nn* | *linear* | *k-nn* |
| SwAV | ✗ | **89.17** | **84.18** | **64.88** | **53.32** |
| | ✓ | 88.79 | 84.01 | 64.21 | 53.07 |
| VICReg | ✗ | **92.07** | **87.38** | **68.54** | **56.32** |
| | ✓ | 91.35 | 86.75 | 67.27 | 55.79 |
| MoCoV3 | ✗ | **93.10** | 89.06 | **68.83** | **58.09** |
| | ✓ | 93.05 | **89.09** | 68.72 | 58.02 |

## C.5. Discussion and Future Work

The relationship between image augmentation and SSL via joint embedding architectures has grown increasingly intricate. Over the past few years, many studies (Chen et al., 2020b; Grill et al., 2020; Wagner et al., 2022; Morningstar et al., 2024) have emphasized the critical role of image augmentation in JEA, demonstrating that making subtle modifications to image augmentations, such as merely adjusting the parameters of *ResizedCrop* and *ColorJitter*, can significantly impact the performance of SSL models. However, recent works (Assran et al., 2023; Moutakanni et al., 2024) have begun to challenge this paradigm by exploring new self-supervised learning frameworks that eliminate the reliance on hand-crafted data augmentations. These efforts argue that specific augmentations may introduce strong biases that could be detrimental to certain downstream tasks (Assran et al., 2022a) and that the most effective augmentations are often task-specific, depending on the domain, rather than adhering to universally hand-crafted settings (Bendidi et al., 2023; Asano et al., 2019; Geiping et al., 2022; Purushwalkam & Gupta, 2020).

Interestingly, Moutakanni et al. (2024) successfully demonstrate that hand-crafted or domain-specific data augmentations are not essential for training state-of-the-art joint embedding architectures when scaling self-supervised learning. Their findings reveal that, with sufficiently large datasets, simple crop of images alone can achieve remarkable results. Notably, this observation aligns perfectly with the characteristics of ReSA. As we show in Section 4.3, removing any single transformation enhances the clustering properties of the representations learned during training, enabling ReSA to better capture the inherent clustering information within the data. When the dataset size is sufficiently large, eliminating all hand-crafted data augmentations perfectly unleashes the innate potential of ReSA. We look forward to future research validating this hypothesis and applying ReSA to large-scale pretraining scenarios.

