# OpenReview forum: "Clustering Properties of Self-Supervised Learning"
_ICML.cc/2025/Conference — ICML 2025 poster_

### Official Review · Reviewer_kGg9 · 2025-03-01

**Overall Recommendation:** 3

**Summary:**

This paper studies the clustering properties of self-supervised learning. This paper finds that the encoder's output exhibits superior and more stable clustering properties than other components. Based on insight, this paper proposes a novel positive feedback method to improve the representation ability further.

The idea of positive-feedback is interesting. Extensive experiments validate the feasibility of this idea.

The finding on the clustering properties of encoding is interesting. However, the underlying reason is not well explained.

**Claims And Evidence:**

The clustering properties of encoding are interesting.
However, I cannot understand why encoding and embedding have different clustering properties. To my knowledge, both embedding and encoding can be regarded as features of neural networks with various depths. So why will adding more layers (from encoding to embedding) degrade the clustering properties?

In Figure 3, how to get the ARI values?

**Essential References Not Discussed:**

No.

**Experimental Designs Or Analyses:**

Yes.

**Methods And Evaluation Criteria:**

Yes.

**Other Comments Or Suggestions:**

No.

**Other Strengths And Weaknesses:**

Strengths:

The experiment is through.

Weakness:
This paper needs to calculate the $A_H$, which seems computationally heavy.

**Questions For Authors:**

The idea of positive feedback is interesting. In line 117, we can see that this paper aims to enhance the clustering structure in the encoding. But my question is, what if an error exists (some samples belonging to different classes are in the same cluster)? Will the proposed method boost this error?

In 139, $A_H$ is a clustering assignment matrix. But in line 146, it becomes a doubly stochastic matrix. The assignment matrix and the stochastic matrix have different sizes.

In Eq. (3), if the encoding replaces the embedding, will Eq. (3) produce better performance?

**Relation To Broader Scientific Literature:**

This paper is related to self-supervised learning.

**Theoretical Claims:**

This paper lacks a theoretical analysis. I suggest the author analyze the theoretical foundations of the findings.

---

> ### Author Rebuttal · Authors · 2025-03-26
>
> Dear Reviewer kGg9,
>
> We sincerely thanks for your time and the valuable suggestions to improve this paper. Here we address each of the comments in detail.
>
> **Why encoding and embedding have different clustering properties**: Thanks for this constructive question. Due to word count limitations, we kindly ask you to refer to our response to Reviewer 69BS, specifically under the section **Why encodings inherently possess better clustering capability**. We sincerely appreciate your understanding and attention to this matter.
>
> **How to get the ARI values**: In Definition A.2 of Appendix A, we provide a detailed definition and formula for the Adjusted Rand Index (ARI). For practical computation, you may also refer to "sklearn.metrics.adjusted_rand_score".
>
> **Theoretical foundations of the findings**: Thanks sincerely for this insightful suggestion. The theoretical exploration of the distinction between encoding and embedding remains a challenge in SSL. We will follow your suggestion and continue to investigate how ReSA facilitates positive feedback learning, optimizes the model's noise robustness, and whether it offers improved bounds compared to other methods.
>
> **$\mathbf{A} _\mathbf{H}$ seems computationally heavy.**: We note that the Sinkhorn-Knopp algorithm, which is used to compute $\mathbf{A} _\mathbf{H}$, does **not** involve gradient propagation and requires only three iterations, making it both efficient and practical as reported in Caron et al. (2020). To quantitatively describe the computation time of the Sinkhorn-Knopp Algorithm, we conducted experiments and found that using an A100-PCIE-40GB GPU, the time for a single computation of a 1024 * 1024 tensor matrix (as used in our experiments) is **0.001578** seconds. This means that when training ResNet-50 on ImageNet for 1 epochs (1251 iterations), the total time for Sinkhorn-Knopp algorithm is about 1.974 seconds. As shown in Table 4, the overall training time is 0.16 × 3600 = 576 seconds, so the proportion of time spent on Sinkhorn-Knopp algorithm is approximately 1.974 / 576, which is about **0.3%**. Moreover, when using larger models, the time spent on the Sinkhorn-Knopp algorithm will represent an even smaller proportion of the total computation time.
>
> [1] Caron et al. (2020). Unsupervised learning of visual features by contrasting cluster assignments.
>
> **What if an error exists (some samples belonging to different classes are in the same cluster)? Will the proposed method boost this error?**: Thank you for this insightful question. In the context of self-supervised learning, the issue of some samples from different classes being assigned to the same cluster is quite likely, especially when there is no label information guiding the process. This is a challenge faced by most clustering-based SSL methods. However, numerous experiments have shown that ReSA exhibits relatively slow performance improvement in the early stages of training, but achieve significantly better results in the later stages compared to other methods, such as contrastive or decorrelation-based SSL, as shown in Figure 9 of our paper and https://anonymous.4open.science/r/long-tailed-evaluation-2843. This suggests that while ReSA may suffer from incorrect cluster assignments in the early training stages, which slows down learning, it is able to gradually address these issues and learn the correct clustering patterns as training progresses, rather than boosting the errors. Furthermore, clustering-based SSL methods like SwAV, DINO and ReSA typically employ clustering assignments with sharp distribution, which help mitigate the impact of incorrect assignments. Investigating the optimization dynamics of clustering-based SSL is an intriguing theoretical direction, and we hope to explore this area in the future.
>
> **The assignment matrix and the stochastic matrix**: We note that $\mathbf{A} _\mathbf{H}$ is the clustering assignment matrix obtained by applying the Sinkhorn-Knopp algorithm to $\mathbf{S} _\mathbf{H}$. The purpose of the Sinkhorn-Knopp algorithm is to ensure that the output is a doubly stochastic matrix, where each row and column sums to 1. Therefore, $\mathbf{A} _\mathbf{H}$ is a doubly stochastic clustering assignment matrix, and the matrices $\mathbf{A} _\mathbf{H}$ on Line 139 and 146 are exactly identical.
>
> **If the encoding replaces the embedding, will Eq. (3) produce better performance?**: Thank you for this thought-provoking question. In Eq. (3), SwAV requires learnable prototypes to perform clustering. If the encoding is used in place of the embedding—namely $\mathbf{Q}=sinkhorn(\mathbf{H}^\top \mathbf{C})$—we must ensure that the encoding and embedding output dimensions match. We tested this approach in early experiments on ImageNet-100 and found that, although training proceeded normally, the resulting performance was quite poor. We speculate that this may be due to the difficulty of effectively optimizing the prototypes under this condition.

---

> > ### Comment · Reviewer_kGg9 · 2025-04-05
> >
> > Thanks for the authors' rebuttal. Part of my concerns have been addressed. The new results on ReSA's learning curves and their comparisons are very interesting. But I still have the following questions.
> >
> > The authors did not explain why ReSA suffers from incorrect cluster assignments in the early training stages, and it can gradually address these issues and learn the correct clustering patterns as training progresses. Actually, the observation is interesting, but the reason behind it is not clear.
> >
> > If the encoding replaces the embedding, can you perform K-means or K-means++ to get the prototype?
> >
> > Besides, the understanding of why encodings inherently possess better clustering capability is still theoretically unclear.
> >
> > Nevertheless, I will maintain my positive score, as the observations on the different behaviours of encoding and embedding are interesting.

---

> > > ### Author Response · Authors · 2025-04-05
> > >
> > > Dear Reviewer kGg9,
> > >
> > > We sincerely appreciate your thoughtful feedback and are truly honored by your kind words regarding the novelty of our findings. It is a privilege to receive your recognition. In response, we would like to offer further clarifications to address the remaining points you raised.
> > >
> > > **Why ReSA can gradually address ...**
> > >
> > > For clarity, we reformulate Equation 2 in the paper to express it in terms of a single sample $z _i$, omitting the symmetric terms.
> > >
> > >  $l _{\text{ReSA}}(z _i) = - \sum _{j=1}^m  {\mathbf{A} _\mathbf{H}}^{(i,j)} \log\frac{e^{z _{i} ^\top z' _{j} /\tau}}{ \sum _{k=1}^m e^{z _{i} ^\top z' _{k} /\tau}}$
> > >
> > > Here, ${\mathbf{A} _\mathbf{H}}^{(i,j)}$ denotes the $ (i,j) $-th element of $ \mathbf{A} _\mathbf{H} $, where $ \mathbf{A} _\mathbf{H} = \text{Sinkhorn}(\mathbf{H}^\top \mathbf{H}) $. Given that the Sinkhorn-Knopp algorithm exhibits strict monotonicity, we have the property that if $ {h _i}^\top {h _j} > {h _i}^\top {h _k}$, then $ {\mathbf{A} _\mathbf{H}}^{(i,j)} > {\mathbf{A} _\mathbf{H}}^{(i,k)}$.  Since the vectors in $ \mathbf{H} $ are $\ell _2 $-normalized, the diagonal elements of $\mathbf{H}^\top \mathbf{H}$ are all maximized to 1. This implies that for any $ i $ and $j $, we have $ {\mathbf{A} _\mathbf{H}}^{(i,i)} \geq {\mathbf{A} _\mathbf{H}}^{(i,j)} $. Furthermore, due to the sharp distribution employed in Sinkhorn, the diagonal elements are significantly larger than the off-diagonal elements. This ensures that the optimization focus of ReSA remains on the alignment of positive samples, substantially reducing the impact of early assignment errors (off-diagonal elements) on the training process. Throughout training, the continual alignment of positive samples enables the model to learn meaningful representations, which in turn facilitates correct cluster assignments in the later stages, further promoting learning.
> > >
> > >  In contrast, other clustering-based methods such as SwAV and DINO require an initialized prototype for cluster assignment, making it more challenging to avoid early clustering errors. We speculate that this is one of the key reasons why ReSA outperforms these methods in terms of overall effectiveness.
> > >
> > > **Can you perform K-means or K-means++ to get the prototype?**
> > >
> > > We are both surprised and honored to find that your insights align so closely with the research trajectory we have followed in this work. At the onset of this project, our initial approach was to use k-means for calculating hard pseudo-labels in the encoding phase, and then apply these pseudo-labels to compute the supervised contrastive loss  [1]. We also explored the use of k-means++ and soft k-means, and the results we obtained from training on CIFAR-10 are as follows:
> > >
> > > | method   |   knn acc | linear acc  |
> > > | :-------- | --------:| :--: |
> > > |  k-means  | 89.72 | 91.95  |
> > > |  k-means ++ |   90.09 |  92.35  |
> > > |  soft k-means | 90.17 | 92.41  |
> > > |  ReSA | 93.02 |  93.53  |
> > >
> > >  [1] Khosla P, et al. Supervised contrastive learning.
> > >
> > >  Similar to DeepCluster, using k-means to calculate pseudo-labels requires storing the encoding vectors for the entire dataset (as the performance of mini-batch k-means is poor), and performing k-means on the entire dataset is computationally intensive. Consequently, we abandoned this approach and, in subsequent work, introduced ReSA, which outperforms existing state-of-the-art methods in both performance and training efficiency.
> > >
> > > **the understanding of why encodings ... is still theoretically unclear.**
> > >
> > > Although the theoretical investigation of the projector remains an open research challenge within the community, we are actively working towards addressing this issue. Given Intra-Class Compactness $ \mathcal{D} _{\text{intra}}^{(c)}(\mathbf{H}) = \mathbb{E} _{x _i, x _j \sim c}  \||h _i - h _j\||^2 $ and Inter-Class Separability $  \mathcal{D} _{\text{inter}}^{(c _1, c _2)}(\mathbf{H}) = \mathbb{E} _{x _i \sim c _1, x _j \sim c _2}  \||h _i - h _j\||^2  $, we define the clustering ratio  $ \mathcal{R}(\mathbf{H}) = \frac{\mathbb{E} _c [\mathcal{D} _{\text{intra}}^{(c)}(\mathbf{H})]}{\mathbb{E} _{c _1 \neq c _2} [\mathcal{D} _{\text{inter}}^{(c _1, c _2)}(\mathbf{H})]} $  and $ \mathcal{R}(\mathbf{Z}) = \frac{\mathbb{E} _c [\mathcal{D} _{\text{intra}}^{(c)}(\mathbf{Z})]}{\mathbb{E} _{c _1 \neq c _2} [\mathcal{D} _{\text{inter}}^{(c _1, c _2)}(\mathbf{Z})]} $, where a lower $\mathcal{R} $ indicates better clustering property.  Under InfoNCE optimization, we are exploring the theoretical relationship between $\Delta \mathcal{R}(\mathbf{H}) $ and $ \Delta \mathcal{R}(\mathbf{Z}) $ during training, where $ \Delta \mathcal{R}$ denotes the improvements in the clustering ratio after one optimization step. This relationship may not be fixed, as our experiments have shown that the clustering performance of the embeddings improves more rapidly in the early stages of training, but tends to decline in the later stages.

---

### Official Review · Reviewer_PEEd · 2025-03-12

**Overall Recommendation:** 4

**Summary:**

The paper presents a new self-supervised learning method that encourages clustering in the output "embedding" layer based on clustering found in the earlier "encoding" layer used as representations for downstream tasks.

The paper analyses clustering metrics on representations extracted at different layers of the NN architecture (encoder + projector), to identify where clustering might be better applied and thereby derive a training objective that combines concepts used previously in SSL in a new, arguably simpler way.

**Claims And Evidence:**

The paper claims to achieve superior performance to existing methods, which seems well justified, subject to including all relevant comparators. (see below)

**Essential References Not Discussed:**

See "Relation to Broader.." (above)
- MUGS and MSN (https://arxiv.org/pdf/2204.07141, https://arxiv.org/pdf/2203.14415)
- Bizeul et al. (https://arxiv.org/pdf/2402.01399)

**Experimental Designs Or Analyses:**

The protocol seems standard for SSL papers

**Methods And Evaluation Criteria:**

The method is based on appropriate analysis (of clustering in NN layers) and applies clustering, used extensively in prior SSL works, in a novel way that makes intuitive sense.

**Other Comments Or Suggestions:**

087R: The citation is DeepCluster, text says DeepCluster2?

**Other Strengths And Weaknesses:**

* Well written and clear paper.
* Prepared to raise score if concerns are appropriately addressed.

**Questions For Authors:**

* 086L: Do you train 1 neural network function or 2 independent functions with distinct parameters? This gives the impression of 2, but later it seems only 1. If 1, then this presentation is misleading and should be made more clear and consistent with the paper and model that follows. This is important for a clear understanding of how representations are a function of the data.
* 083R: "it remains unclear.." - Why would this *not* continue through all layers of the network (a layer doesn't "know" which part its in)?
* 085R: "embedding tends to be less influenced by clustering irrelevant features" - what does this mean?
* 100R: what is clustering "capability" and "stability" in local clustering? (e.g. stability with respect to what?)
* 142L: "These include excellent local clustering ability .. and stability ..., global clustering capability and similarity measure effectiveness (ARI)." Can you put this in clearer every day language?   What is the "ability" of a set of vectors, etc?
* 208L: $S_M$ for a matrix $M=H$ is already defined (137R) to be symmetric so taking transpose in Eq 2 appears to make no sense. But you then ***re**define* $S_M$ below (for $M=Z$).
    - It would be clearer to explicitly put $Z^TZ'$ in Eq 2, which is then directly comparable to Eq 3.
* 208L: Since $S_H$ is symmetric, is $A_H$ not symmetric also? If so, why transpose? (confusing)
* 214L: no need to define the same function twice for different arguments (confusing)
* 214L: what is the normalisation constant? Is it not the sum over all of S?

**Relation To Broader Scientific Literature:**

* (025R) Caron et al. (2018) should be cited here as one of the first to do this with modern NNs
* (029R) the introduction links clustering and representation learning empirically and theoretically (029R-038R), the latter should include Bizeul et al. (https://arxiv.org/pdf/2402.01399), which proposes a latent variable model that unifies various SSL methods, including clustering methods so seems directly relevant.
  - The clustering nature of ReSA and close relationship shown to InfoNCE suggest ReSA appears consistent with their model. Discussion of this would improve the paper, perhaps adding theoretical justification for why ReSA works/what is implicitly assumed about the data.
* (037L) "Representation Learning with Contrastive Predictive Coding" van den Oord et al. (InfoNCE)  would seem appropriate to cite here as it was a key initiator of recent interest in SSL models.
* Recent works: MuGS and MSN (https://arxiv.org/pdf/2204.07141, https://arxiv.org/pdf/2203.14415) are not referenced or compared to.
    - Please discuss these and reconcile their results, e.g. for ImageNet, with those in the paper.

**Theoretical Claims:**

The paper includes no theoretical claims.

The paper would be improved if it considered how the proposed training objective fits with current theoretical rationale for SSL (see below).

---

> ### Author Rebuttal · Authors · 2025-03-26
>
> Dear Reviewer PEEd,
>
> We would like to begin by sincerely thanking you for the insightful and valuable comments. Below, we address each of the comments in detail.
>
> **Relation to Broader...**:
> We have revisited these papers, and will cite and discuss them in the new manuscript.
> - We will further investigate the ELBO proposed by Bizeul et al. which may be helpful to derive ReSA's bounds.
> - MuGS and MSN are both excellent works that aim to enhance SSL performance in ViTs. In contrast, this work is grounded in traditional SSL benchmarks to demonstrate the effectiveness of encoding-based positive feedback SSL, which is why our experiments primarily use ResNet models. In the past two months, we also explored the effectiveness of ReSA pretraining in ViTs using framework of DINOv2, achieving **77.8%** k-NN accuracy with ViT-B after just 100 epochs, compared to DINOv2's **76.0%** after 100 epochs and MuGS’s **78.0%** after 400 epochs.
>
> **087R**:  Here, we refer to SSL methods that apply clustering constraints on the embedding (i.e., the output of the projector). Since DeepCluster (Caron et al., 2018) does not include a projector, this limitation was addressed and improved as DeepCluster-v2 noted in the SwAV (Caron et al., 2020). We will clarify this point.
>
> **086L**: In the definition of joint embedding architectures (JEA), it is feasible to use either two identical (i.e., one shared) or different neural networks. In practice, using the same architecture without sharing parameters (i.e. the teacher-student distillation with momentum updates as used in BYOL, DINO, and MoCoV3) often yields the best performance. For generality, we do not emphasize the detailed choices of neural networks here, as different methods may adopt different strategies.
>
> **083R**: Thanks again for this constructive question. Due to word count limitations, we kindly ask you to refer to our response to Reviewer 69BS, specifically under the section **Why encodings inherently possess better clustering capability**. We sincerely appreciate your understanding and attention to this matter.
>
> **085R**: As we clarify above, the invariance constraint imposed by the SSL loss can cause the embedding to lose diverse information, which may include clustering-irrelevant features such as background information, color redundancy, viewpoint variations and so on. In such cases, fewer clustering-irrelevant features contribute to better clustering performance. However, the invariance constraint may also lead to the loss of class-relevant information, making it difficult for us to determine which—encoding or embedding—exhibits better clustering properties in this context.
>
> **100R**:
> - Local clustering capability refers to how well the representation partitions the data into meaningful local clusters. A higher mean silhouette coefficient for the representation indicates better separation and tighter grouping of data points within their respective clusters, which demonstrates stronger clustering capability.
> - Local clustering stability is quantified by the standard deviation of the silhouette coefficient. A lower standard deviation signifies that the clustering results are more consistent across the data points, meaning fewer outliers and more stable cluster assignments. This indicates that the clustering representation is robust and less sensitive to variations or noise in the data. We would clarify these more clearly in appendix A.
>
> **142L**: When we talk about the 'ability' of a set of vectors, we are referring to how well they can represent the underlying structure of the data. In this case, 'Global clustering capability' looks at how well the entire set of data is grouped into clusters as a whole, considering the overall structure of the data. As we use k-mean when calculating ARI, so a higher ARI value also indicates that the relationships between the vectors can be well captured and measured using the similarity.
>
> **208L**:
> - We sincerely appreciate your valuable feedback. We will explicitly revise $\mathbf{S} _\mathbf{Z}$ to $\mathbf{Z}^\top \mathbf{Z}'$ for better clarity.
> - Although $\mathbf{S} _\mathbf{H}$ is a symmetric matrix, the iterative row and column normalization inside the Sinkhorn-Knopp algorithm does not preserve symmetry. This is because each row and column can receive different normalization factors during each iteration. As a result, even a symmetric input matrix can become asymmetric after being processed by the Sinkhorn updates.
>
> **214L**:
> - Yes, they indeed represent the same softmax functions. Here we stated them twice to emphasize that these distributions are respectively computed row-wise and column-wise. We appreciate your feedback and will revise it to convey this more clearly.
> - $\mathbf{1} _m \in \mathbb{R}^{m\times 1}$ stands for all-ones column vector as we note in (144R) of Algorithm 1. Here, we express the softmax functions in the form of matrix multiplication, where the denominator corresponds to the sum of each row.

---

> > ### Comment · Reviewer_PEEd · 2025-04-06
> >
> > *Assuming* the paper is improved in line with the above responses I am more in favour of publication, so **increase my score 3 to 4**.
> >
> > In particular though, for ML to "mature" there needs to be far less (if any) divide between "empirical" and "theoretical" works. Neural networks are, of course, popular due to their empirical results and much research remains largely validated empirically, but as theoretical understanding develops, it is important that it is not overlooked and new works should discuss how they fit with it (or not). This paper presents a relatively "simple" model (by which I mean neat, not trivial) and would be more impactful if there were discussion as to *why* it works and what theoretic models of SSL it supports, or otherwise. I note that lack of theoretic grounding is a concern for all reviewers. **My increased rating very much assumes that this point especially is incorporated**, which I would ideally review but since the process does not allow that, I take on faith.
> >
> > Detailed points:
> > * models like MUGS should at least be mentioned as ML is more about the task than the approach, it is completely reasonable to say that the proposed method improves a particular family of models, but should not give the impression of absolute state-of-art if not the case
> > * 085R - this information loss would seem to relate to what is predicted theoretically and shown empirically in Bizeul et al.
> > * 086L - the paper should clearly state what is actually done. A generalisation that is not empirically validated may not be relevant here even if it is elsewhere (e.g. I *doubt* that training two networks would be worthwhile, but either way that would need to be empirically verified).

---

> > > ### Author Response · Authors · 2025-04-06
> > >
> > > Dear Reviewer PEEd,
> > >
> > > We sincerely appreciate your response and are truly grateful for your recognition and guidance.
> > >
> > > We wholeheartedly acknowledge the significance of bridging the gap between empirical research and theoretical foundations. We agree that a more profound theoretical understanding is essential for the advancement of the ML field.
> > >
> > > In our revised manuscript, we will make sure to explicitly highlight how the proposed method aligns with established theoretical frameworks in SSL. Additionally, we will include a detailed discussion on the underlying reasons for the model's effectiveness and the theoretical models it supports, thereby enriching the overall contribution of the paper.
> > >
> > > We value your increased rating and will incorporate the key points in the revised version, which we hope could meet your expectations. In response to your detailed points,
> > >
> > > - we will provide a more thorough comparison of ReSA with other prominent SSLmodels, such as MuGS, and clearly highlight its advantages as well as potential limitations.
> > > - Furthermore, building upon the ELBO proposed by Bizeul et al., we will delve deeper into the theoretical reasons behind the information loss induced by the invariance constraints in SSL, offering a more comprehensive analysis.
> > > - We will provide a clearer and more detailed definition for the joint embedding architectures (JEA). The experiments of training two separate networks was tested in Table 5 of the VICReg paper [1] , but it proves to be not worthwhile, as it yields worse performance compared to training a single network.
> > >
> > > On behalf of all the authors, we would like to express our sincerest gratitude once again and wish you all the best.
> > >
> > > [1] Adrien Bardes, Jean Ponce and Yann LeCun. VICReg: Variance-Invariance-Covariance Regularization For Self-Supervised Learning.

---

### Official Review · Reviewer_69BS · 2025-03-13

**Overall Recommendation:** 3

**Summary:**

This paper investigates the clustering properties inherent in self-supervised learning (SSL) through joint embedding architectures. The authors empirically demonstrate that encoder outputs (encodings) exhibit superior clustering quality compared to projector embeddings, as measured by silhouette coefficients and adjusted Rand indices. Building on this insight, They propose Representation Soft Assignment (ReSA), a novel SSL framework that leverages online self-clustering on encodings to guide representation learning through soft assignment targets. ReSA doesn't require learnable prototypes and executes Sinkhorn-Knopp clustering only once per iteration, achieving computational efficiency while preserving semantic structure.   A weak augmentation strategy that balances clustering stability and invariance learning further enhances the method's effectiveness.

**Claims And Evidence:**

Some important claims lack justification:
1. The core premise "encodings inherently have better clustering properties"—is presented as an empirical observation (Sec 3) but lacks theoretical grounding. The paper empirically demonstrates that encoder outputs (H) exhibit stronger clustering properties than projector embeddings (Z) via metrics like SC/ARI (Fig 3-4). However, no theoretical analysis explains why encodings inherently possess better clustering capability. Does the encoder’s position (pre-projector) naturally filter augmentation noise or preserve semantic information? It is suggested to quantify the semantic preservation difference of H/Z through methods such as information bottleneck theory.
2. There is an error propagation risk that exists: The positive feedback of ReSA depends on the quality of cluster assignment, If the initial clusters are noisy (such as in the early stages of training), the doubly stochastic matrix AH reinforces incorrect associations.
While Figure 11 suggests that Sinkhorn-Knopp mitigates early noise, there is no proof that the ReSA loss function suppresses error propagation. The paper lacks quantitative evidence that ReSA robustly handles noisy clusters. No analysis shows whether initial noisy assignments (e.g., low-ARI epochs) get amplified via the feedback loop.

**Essential References Not Discussed:**

No.

**Experimental Designs Or Analyses:**

Experiments only use balanced datasets (e.g., CIFAR, ImageNet). There is no validation on long-tailed or domain-shifted data. It is suggested that test ReSA under imbalanced data distributions** to assess whether clustering-guided learning amplifies biases.

**Methods And Evaluation Criteria:**

In the section "3. Exploring Clustering Properties of SSL"， the evaluation exclusively focuses on SSL methods with projector heads, omitting architectures without projectors. This introduces selection bias, as the observed encoding superiority (Sec 3) might be contingent on projector-augmented frameworks rather than intrinsic to encodings.
Suggested Improvement: Validate clustering metrics on projector-free SSL baselines to disentangle whether the encoding advantage stems from architectural constraints or fundamental representation properties.

**Other Comments Or Suggestions:**

It is recommended that the section “5.3. Transfer to Downstream Tasks” be more detailed, as this is an advantage of SSL to better assess the validity of the ReSA methodology proposed in this paper.

**Other Strengths And Weaknesses:**

1. The theoretical basis of this paper is insufficient, for the core premise-“encodings inherently have better clustering properties”, even though the authors try to prove it from an experimental point of view, I still think the experimental results are not significant enough.
2. The paper is well presented in terms of iconography and text, with a good summary of previous work and clear claims.
3. The innovation of this paper is okay, and the use of soft assignment does solve the problem that samples belonging to the same category may be unintentionally pushed farther away during the training process, thus destroying the underlying semantic clustering structure, and making full use of the hidden information contained in the encoding. It would be an excellent work if the authors could add the theoretical basis for the important points in the paper and optimize the noise robustness of the model.

**Questions For Authors:**

See the weaknesses.

**Relation To Broader Scientific Literature:**

The paper is highly related to these deep clustering methods based on SSL such as contrastive clustering.

**Theoretical Claims:**

The paper does not present formal theoretical proofs so it is suggested that providing the theoretical proof of the core premise—"encodings inherently have better clustering properties"

---

> ### Author Rebuttal · Authors · 2025-03-26
>
> Dear Reviewer 69BS,
>
> We sincerely thanks for your time and the valuable feedback for this paper.
>
> **Why encodings inherently possess better clustering capability**:
> We would like to thank the reviewers for their insightful discussion regarding the distinction between encoding and embedding. To the best of our knowledge, the projector has become an indispensable component of JEA-based SSL. However, the dynamics of its optimization and the reasons behind its success remain an open question within the community. Some works have attempted to explain these principles. For example, Jing et al. (2022) [1] empirically discovered that applying SSL loss either to encoding or embedding led to a significant decrease in the rank of the corresponding features. They argued that this rank reduction indicates a loss of diverse information, which, in turn, reduces generalization capability. This explanation aligns with the hypothesis in SimCLR [2], **where the additional projector acts as a buffer to prevent information degradation of the encoding caused by the invariance constraint**. Additionally, Gupta et al. (2022) [3] 's null space analysis for the projector posited that the projector might implicitly learn to select a subspace of the encoding, which is then mapped into the embedding. In this way, only a subspace of the encoding is encouraged to be style-invariant, while the other subspace can retain more useful information.
>
> Therefore, the SSL constraint can cause the embedding to lose information, which may include not only clustering-irrelevant features such as background information, but also class-relevant information, making it difficult for us to determine which—encoding or embedding—exhibits better clustering properties in this context. Based on this, our paper empirically demonstrates that the projector indeed filters out information that could support clustering, while encoding achieves better clustering performance.
>
> **theoretical analysis on ReSA**: We are truly grateful for your expectations and support for this work. As clarified above, the theoretical investigation of the projector remains an open research challenge. The community currently lacks definitive theoretical evidence explaining why the encoding can achieve superior downstream performance compared to the embedding. Building on your insights, we can first conduct interpretability experiments to examine which class-relevant and class-irrelevant features are captured by both the encoding and the embedding, thereby demonstrating how the encoding may possess better clustering properties. We also appreciate your suggestion to quantify this semantic preservation difference through methods such as information bottleneck theory. This could be a valuable approach to theoretically investigate why encodings, as opposed to projector embeddings, may lead to better clustering. We will look into incorporating this analysis in the revised version of the paper to provide a more comprehensive theoretical foundation.
>
> **error risk**: We agree that the paper could benefit from a more rigorous analysis of how the ReSA loss function handles error propagation. In the context of self-supervised learning, early clustering errors are inevitable, as there is no label information or other prior knowledge to guide the process. This is a challenge faced by most clustering-based SSL methods. However, numerous experiments have shown that ReSA exhibits relatively slow performance improvement in the early stages of training, but achieve significantly better results in the later stages compared to other methods, as shown in Figure 9 of our paper and https://anonymous.4open.science/r/long-tailed-evaluation-2843. This suggests that while ReSA may suffer from incorrect cluster assignments in the early training stages, which slows down learning, it is able to gradually suppresses error propagation and learn the correct clustering patterns as training progresses, rather than amplifying the errors.
>
> **projector-free SSL baselines**: To the best of our knowledge, the projector has become an indispensable component of JEA-based SSL. Traditional projector-free baselines, such as the MoCo (He et al. 2020), no longer achieve competitive results, while newer versions like MoCo V2 and V3 have demonstrated the projector’s crucial role in enhancing representational generalization.
>
> **long-tailed data**: Dear Reviewer, please refer to https://anonymous.4open.science/r/long-tailed-evaluation-2843.
>
> **Transfer to Downstream Tasks**: In addition to the experiments described in Section 5.3, we also perform transfer learning on fine-grained datasets (Table 6) and conduct a low-shot evaluation (Table 10). We will include these details in Section 5.3 as well.
>
> [1] Understanding dimensional collapse in contrastive self-supervised learning.
>
> [2] A Simple Framework for Contrastive Learning of Visual Representation.
>
> [3] Understanding and Improving the Role of Projection Head in Self-Supervised Learning.

---

### Decision · Program_Chairs · 2025-05-01

**Decision:**

Accept (poster)

**Comment:**

This work identifies an interesting observation: encoder outputs in joint embedding architectures display stronger and more stable clustering properties than projector embeddings. Building on this insight, the authors propose a new self-supervised learning method, ReSA, which introduces a positive-feedback mechanism to guide learning via soft clustering assignments. ReSA avoids the use of learnable prototypes, relies on a computationally efficient Sinkhorn-Knopp-based formulation, and achieves consistently strong results across standard SSL benchmarks.

While several reviewers initially expressed reservations about the lack of theoretical analysis and the potential risks of reinforcing early clustering noise, the authors provided thoughtful and comprehensive responses. They clarified the empirical basis for the encoder's superior clustering properties and outlined relevant directions for developing theoretical support in future work. They also presented evidence suggesting that the proposed method is robust to early-stage noise, gradually improving clustering quality as training progresses. Additional clarifications regarding the model architecture, efficiency of the Sinkhorn step, and practical advantages over baseline clustering methods were well-received by the reviewers.

Following the rebuttal, all reviewers either maintained or increased their scores, and acknowledged that their key concerns had been addressed. The paper presents a novel and well-motivated approach that is both simple and effective, and is likely to be of interest to the community studying SSL and clustering-based learning dynamics.